# Robust triboelectric energy harvesters engineered from electrochemically deposited films of HKUST-1 polycrystals
Chuzhan Jin & Jin-Chong Tan ✉

Triboelectric nanogenerators (TENGs) offer a potential power source for compact electronics and self-powered portable devices, with increasing interest in integrating metal-organic frameworks (MOFs) due to their tunable physical and chemical properties. However, the direct incorporation of MOF powders in TENG is hindered by their weak and unstable attachment to the underlying conductive substrates. Herein, we show that electrochemical MOF deposition offers a facile approach to deposit hydrophilic HKUST-1 films on a copper electrode, yielding a robust tribopositive layer after a growth time of 2 h. When this surface was impacted against a tribonegative layer such as Kapton under contact-separation mode, the optimal output of the TENG device reached the highest voltage output of ~99 V with a power density of $771.8 \pm 0.3$ mW m$^{-2}$. The device exhibits extended stability, with a negligible voltage decay under ambient environment and exposed to relative humidity from 10% to 70%. This study demonstrates a feasible strategy to generate mechanically resilient MOF-based TENGs with reproducible output for real-world environmental conditions.

As the demand for portable electronics continues to grow, the development of sustainable power sources and compact energy harvesters has become increasingly pertinent. Harnessing ambient mechanical energy from everyday movements, surrounding acoustic vibrations, and even ocean waves gives a pathway to power small-scale electronics while contributing to net-zero emission goals. Triboelectric nanogenerator (TENG) represents an innovative breakthrough towards this aspiration, as first proposed by Wang et al. in 2012, TENG offers a simple and cost-effective solution for converting environmental mechanical energy into electrical power by harnessing triboelectrification and electrostatic induction[1]. Recent advancements have positioned TENGs as a promising renewable energy solution with wide-ranging applications from energy generation to self-powered sensors[2]. To date, TENGs have been developed in four operational modes: contact-separation mode[3], lateral-sliding mode[4], single-electrode mode[5], and free-standing mode[6]. Among these modes, the contact-separation mode is the most widely studied, requiring two dielectric materials or electrodes with significant differences in electron affinity, enabling strong electron transfer and charge accumulation. Thus, material selection and design play a crucial role in optimising future TENG performance[7]. The first triboelectric series, proposed in 1757 and later refined by AlphaLab Inc. in 2009, primarily included polymers and metals, thereby limiting the scope of materials studied for TENG applications[8]. Recent advancements have expanded the range of materials used in TENGs, including TiO$_2$ nanotubes[9], graphene[10],

and perovskites[11], all of which can be promising candidates for TENG. However, one class of materials, metal-organic frameworks (MOFs), is relatively unexplored in the context of TENG technology.

MOFs are porous crystalline compounds composed of metal ions or clusters linked by organic ligands, forming a three-dimensional (3D) network[12]. Their designable structures can be chemically modified, and due to their large surface area, tunable pore structures, and high crystallinity, MOFs are increasingly being explored for applications in TENGs[13]. Most research in this area has focused on preparing composite films by mixing MOF particles with organic polymers to harness the combined properties of both materials[14,15]. For instance, Wang et al. developed a TENG using UiO-66-4F, a MOF with strong electron-withdrawing groups incorporated into a PDMS matrix. The resulting MOF–polymer composite film achieved an output voltage of 937 V and a power density of 38.7 W m$^{-2}$ [16]. However, polymer integration often masks the intrinsic properties of the MOF itself. Khandelwal et al. first reported the direct use of MOFs in TENGs by growing ZIF-8 on an indium tin oxide-coated polyethylene terephthalate (ITO-PET) substrate[17]. This method produced a high-performance TENG with an output voltage of 164 V and power density of 0.39 W m$^{-2}$. Following this, Khandelwal et al. expanded their investigations to include other MOFs, such as ZIF-7, ZIF-9, ZIF-11, ZIF-12, ZIF-62, and MIL-88A, adhering them to aluminium (Al) tape to assess their triboelectric properties[18,19]. Babu et al. used a hydrothermal method at 70 °C to directly grow ZIF-67 on Al foil,

Multifunctional Materials and Composites (MMC) Laboratory, Department of Engineering Science, University of Oxford, Oxford, UK.
✉e-mail: jin-chong.tan@eng.ox.ac.uk

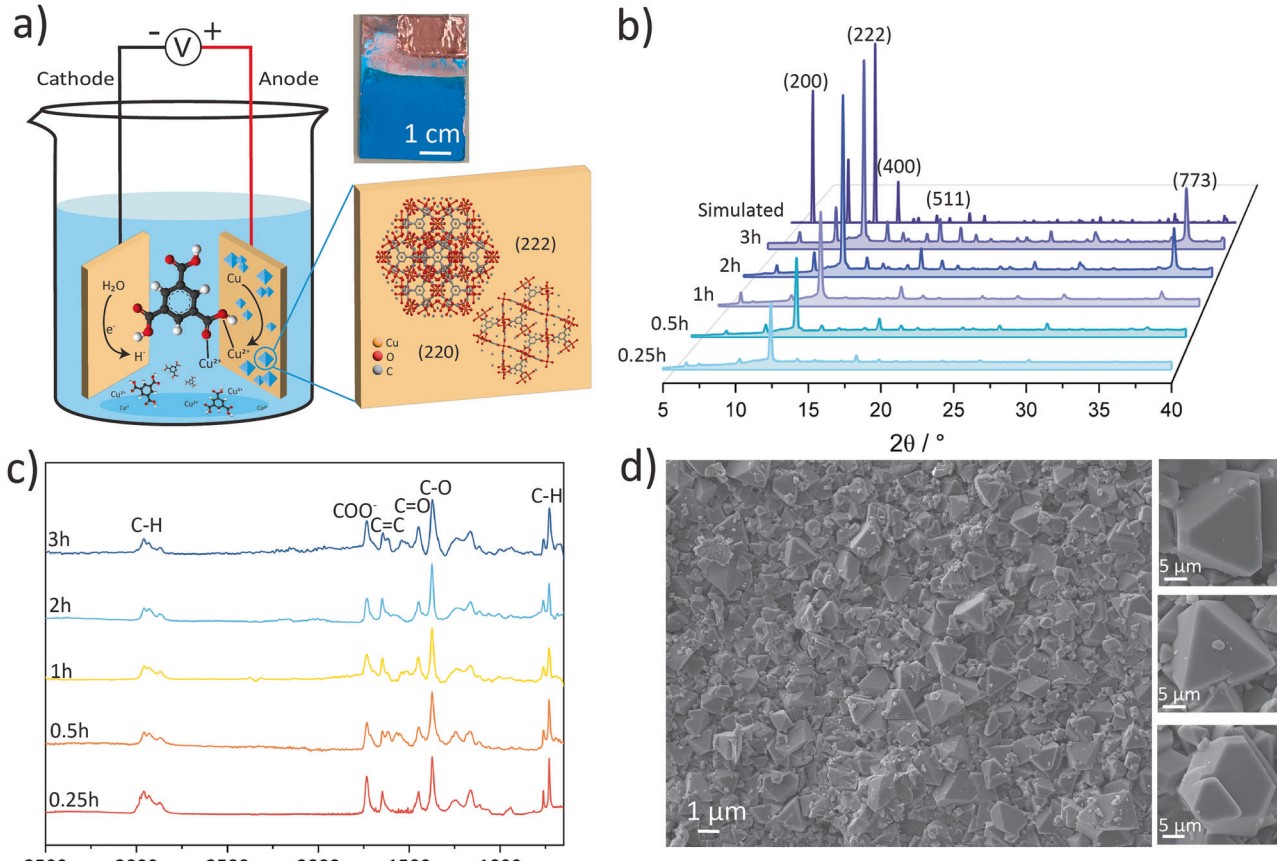

**Fig. 1 | Synthesis and characterisation of HKUST-1. a** Schematic diagram of anodic electrochemical synthesis of HKUST-1 on a copper base substrate, with an inset photo of HKUST-1 polycrystalline film deposited after 2 h growth time and representation of the two primary crystalline facets oriented at (222) and (220). **b** XRD patterns of the as-synthesised HKUST-1 film compared with the simulated pattern (CCDC code FIQCEN). **c** FTIR spectra of as-synthesised HKUST-1 as a function of electrochemical deposition time. **d** SEM micrographs of the 2 h-HKUST-1 sample showing the (222) facets facing upward on the Cu plate.

achieving a power density of 2.4 W m$^{-2}$ [20]. Slater et al. further explored the ZIF series by simply adhering MOF powder to Al tape as a supporting substrate, though these methods suffer from weak interfacial adhesion, compromising crystal stability on the substrate[21].

Despite extensive research on the ZIF series, other MOFs, such as HKUST-1 ($Cu_3(BTC)_2$), have been less explored for their triboelectric properties. Wen et al.-doped HKUST-1 into PDMS and found that a 5 wt% loading of MOF filler achieved an output of 205 V and a power density of 7.9 W m$^{-2}$ under 10% relative humidity (RH), with even higher performance observed at up to 90 RH%[22], which is due to the higher dielectric constant associated with greater density and polar groups such as $COO^-$[14,23]. The pure HKUST-1 crystal has a dielectric constant of 2.95 at 1 MHz frequency in the (100) direction, which is higher than most ZIF crystals[24,25]. Thus, it is important to study the triboelectric performance of intrinsically grown HKUST-1 film with a strong attachment to the TENG substrate, which is the focus of our study.

HKUST-1 ($Cu_3(BTC)_2$, where BTC = 1,3,5-benzenetricarboxylate or trimesate) is a $Cu^{2+}$-centred 3D MOF material known for its large surface area, pore size, and hydrophilic nature[26–28], which can be readily produced through anodic electrodeposition, a method patented by BASF in 2005[29]. Previously, Buchan et al. investigated the mechanical properties of HKUST-1 polycrystalline films deposited on Cu plates via an electrochemical reaction, demonstrating that adhesion strength can be controlled by adjusting the thickness of HKUST-1 and the roughness of substrates[30]. Building on this research, the current study employed the electrochemical method to both stabilise the crystal and develop a device for measuring the triboelectric response of HKUST-1. The primary advantages of the electrochemical

method include its relatively fast synthesis at lower temperatures, mild synthetic conditions, short crystal growth times, and scalability. In this study, the HKUST-1 films were grown on Cu plates at 55 °C using electrodeposition times of 0.25, 0.5, 1, 2, and 3 h. The triboelectric behaviour of the resultant bluish HKUST-1 functionalised Cu plates (Fig. 1a) was tested against the Kapton tape under the contact-separation mode (Fig. 2a), revealing that the triboelectric properties of HKUST-1 polycrystalline films depend on material growth time and surface morphology. Additionally, the hydrophilic HKUST-1 was evaluated under a RH value of 70%–10%, thereby establishing the influence of humidity on its triboelectric performance as an exemplar of a water-responsive MOF material.

## Results and discussion

### HKUST-1 polycrystalline films

During the electrochemical reaction, upon applying a direct current (DC) potential of 2.5 V, electrons begin to flow from the cathode (Fig. 1a). Given that the electrolyte consisted of a 56:44 ethanol-to-water ratio, two possible reduction reactions can occur at the cathode. However, these reactions require an overpotential to proceed. Therefore, the reduction of water dominates at the cathode due to its lower energy requirement (Eq. (1)). The tributyl methylammonium methyl sulfate (MTBS) in the electrolyte maintains ionic balance and stabilises the electrolyte solution, facilitating efficient ion transport during the electrosynthesis without directly participating in the formation of HKUST-1. The electrosynthesis occurs on the anodic side, where the copper metal is used as a metal source and is oxidised to copper ions ($Cu^{2+}$), dissolving into the electrolyte (Eq. (2)). The ligand in the electrolyte is deprotonated to $BTC^{3-}$, which then reacts with the $Cu^{2+}$

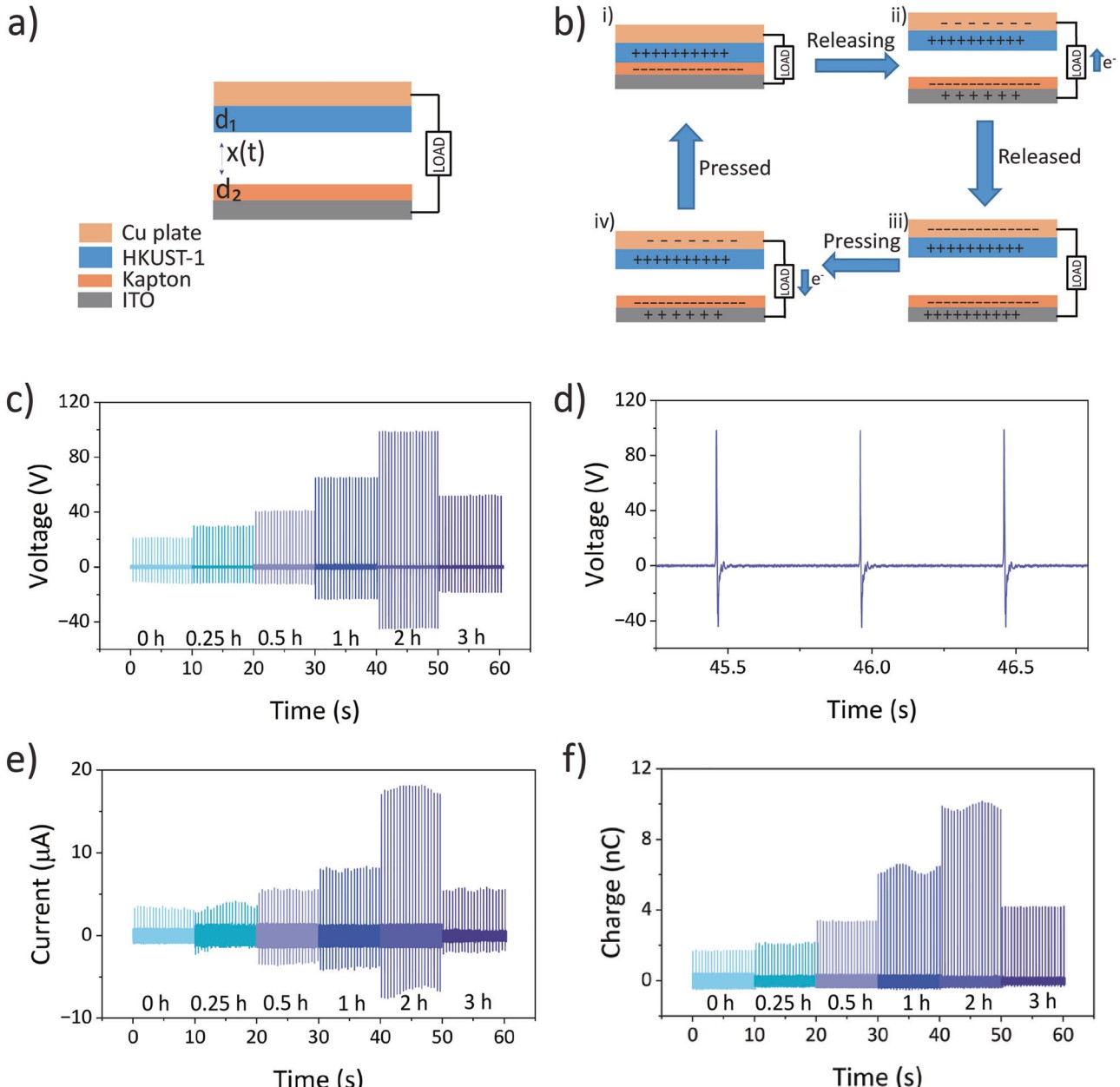

**Fig. 2 | Output performance of HKUST-1-based TENG. a** Schematic diagram of HKUST-1 TENG in contact-separation mode. **b** Working mechanism proposed for the HKUST-1 TENG device under the contact-separation mode against Kapton. **c** Open-circuit voltage of HKUST-1 TENG. **d** Single peak voltage for HKUST-1 sample with a 2 h growth time (extracted from Fig. 2c). **e** Closed-circuit current of HKUST-1. **f** Charge output of HKUST-1.

ions to form the HKUST-1 framework structure, by means of self-assembly on the inner surface of the copper anode (Eq. (3)). The overall reaction is presented in Eq. (4).

$$Cathode : 2H_2O + 2e^- \rightarrow H_{2(g)} + 2OH^-_{(aq)} \tag{1}$$

$$Anode : Cu_{(s)} \rightarrow Cu^{2+}_{(aq)} + 2e^- \tag{2}$$

$$3Cu^{2+} + 2BTC^{3-} \rightarrow Cu_3(BTC)_{2(s)} \tag{3}$$

$$Overall : 3Cu_{(s)} + 2H_3BTC \rightarrow Cu_3(BTC)_{2(s)} + 3H_{2(g)} \tag{4}$$

The as-synthesised HKUST-1 shows a bluish crystalline film covering the Cu plate, with one representative photo shown in the inset of Fig. 1a. The rest of the HKUST-1 sample with other growth times are shown in

Supplementary Fig. 3a–e. The average thickness was found to increase with the processing time, with the exception of the 0.25 h sample due to the incomplete crystal coverage ascribed to a short growth time (Supplementary Table 2), which was calculated from four different locations on the electrode grown with HKUST-1. The HKUST-1 films were characterised using an X-ray diffractometer (XRD) (Fig. 1b). The XRD patterns obtained at various growth times matched the simulated pattern of HKUST-1, confirming the formation of the desired crystalline structure. However, a 0.9° shift was observed in all samples, which we attributed to the thickness effect of the copper substrate (~0.9 mm thickness). The prominent (222) plane at $2\theta = 12°$ in all samples indicated the preferential orientation of the HKUST-1 crystals on the copper plate surface. With the growth time extended, the structural changes could be analysed by the relative peak intensity ratio (Supplementary Fig. 4). The relative diffraction intensities of the (222), (511), and (773) peaks (i.e., $2\theta = 12°$, 18° and 37°, respectively) were analysed as a function of electrodeposition growth time. The increasing ratio from

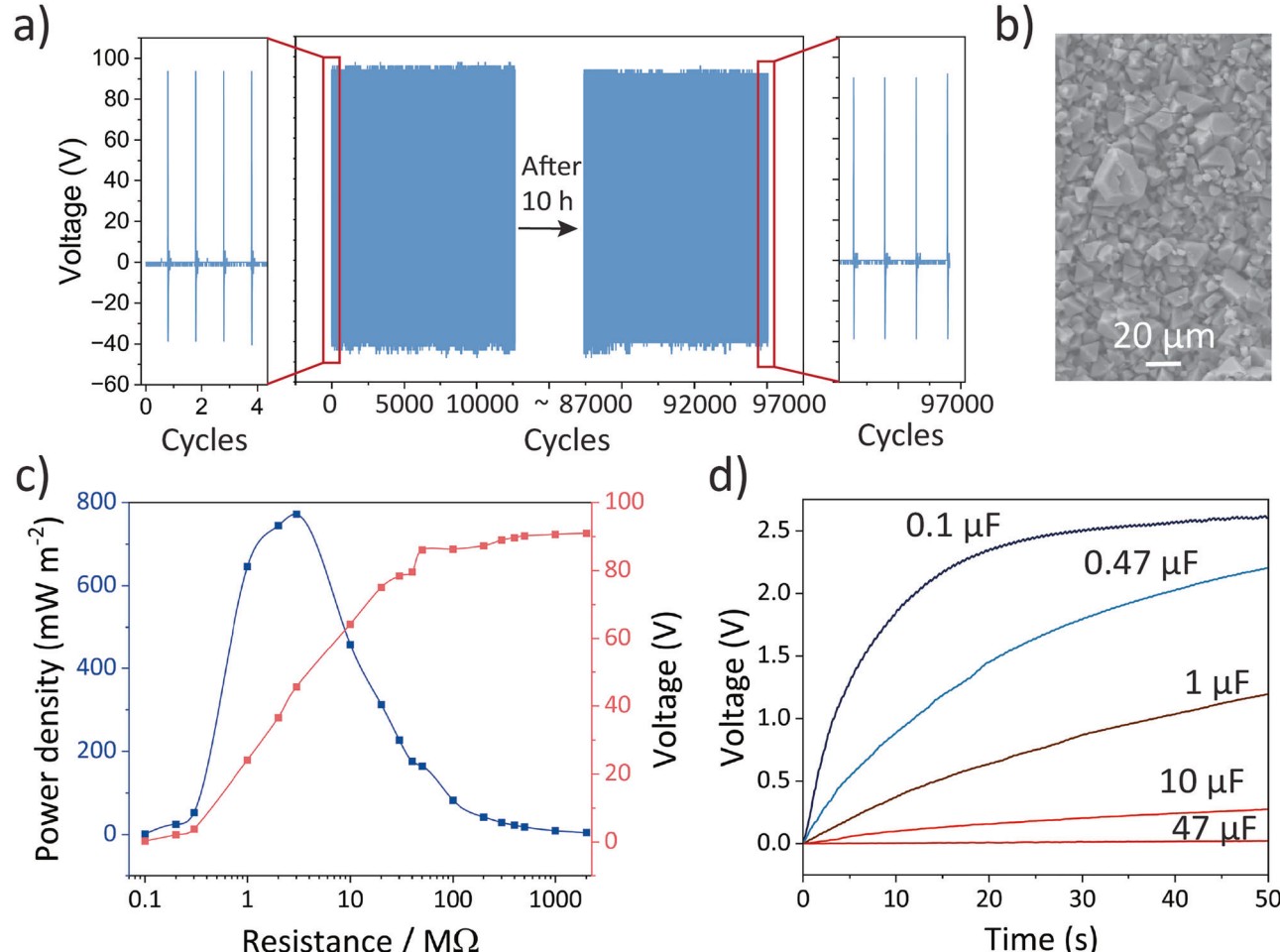

**Fig. 3 | Electrical performance of 2 h-HKUST-1 TENG. a** Durability test of the 2 h HKUST-1 sample over 13.5 h (97,000 continuous impact cycles), with zoomed-in views of the initial and final cycles (exponential decay constant $k$ was estimated in equation ln(95.4/93.6)/97,000, resulting in $k = 1.99 \times 10^{-7}$ cycle$^{-1}$). **b** SEM images of HKUST-1 after the extended contact-separation tests. **c** Power density and voltage output of 2 h-HKUST-1 as a function of load resistance. **d** Capacitor charging curves by operating 2 h-HKUST-1 TENG for 0.1, 0.47, 1, 10 and 47 μF capacitors.

0.25 to 2 h reflected improved crystallinity of HKUST-1 on the copper plate. At 2 h, the reaction was nearly complete, with only a slight increase in the peak intensity ratio observed at 3 h.

Further confirmation of the formation and growth of HKUST-1 was provided by attenuated total reflectance-Fourier-transform infra-red (ATR-FTIR) (Fig. 1c). The ATR-FTIR spectra showed characteristic peaks associated with the $H_3BTC$ organic linker. The vibrational band at around 3000 cm$^{-1}$ was attributed to the C–H stretching mode of the aromatic ring, while the peak near 670 cm$^{-1}$ corresponded to the C–H bending mode. The absorption at 1373 cm$^{-1}$ was assigned to the C–O stretching of the carboxylic groups in $H_3BTC$, and the bands at 1453 and 1544 cm$^{-1}$ were attributed to the C = O stretching. Additionally, the band at 1646 cm$^{-1}$ was assigned to aromatic C = C stretching, and the mode at 1731 cm$^{-1}$ was associated with the COO$^-$ stretching of the BTC linker. SEM characterisation of HKUST-1 crystal morphology (Supplementary Fig. 5) confirmed the orientation determined by XRD analysis, revealing that the (222) facet, characterised by triangular or hexagonal shapes, predominantly appeared on the copper surface. Specifically, Fig. 1d showed that in the 2 h growth sample, most facets attributed to the {111} family of planes were oriented upward, with the three distinctive facet shapes highlighted as examples.

**Performance of HKUST-1 TENG**
The fabrication of the HKUST-1 TENG device was illustrated in Fig. 2a. The HKUST-1 TENG featured a dielectric-to-dielectric interface, where the HKUST-1 crystal plate acted as the tribopositive side, while the Kapton tape served as the tribonegative side. The proposed mechanism of HKUST-1 TENG operating in contact-separation mode is illustrated in Fig. 2b[31]. In the initial stage (Fig. 2a), there is no contact between the layers, so no electron transfer occurs. When the top HKUST-1 layer comes into contact with the Kapton layer due to application of an external compressive force, equal amounts of positive and negative charges are induced on the surfaces of both layers through triboelectrification (Fig. 2b.i)[32]. Once the external force is removed, the potential difference drives the flow of electrons from the Kapton electrode to the Cu electrode through the external circuit (Fig. 2b.ii). Equilibrium is reached when the two layers are fully separated and far apart (Fig. 2b.iii). During the reverse process, when the layers are pressed together again, electron flow occurs in the opposite direction (Fig. 2b.iv). This cyclic process is periodically repeated, thereby generating an alternating electrical output (positive and negative peaks) within each contact-separation cycle (Fig. 2d).

Figure 2c, e and f showed the output performance in terms of the open-circuit voltage ($V_{oc}$), closed-circuit current, and charge output, with an applied external force of around 90 ± 10 N (Supplementary Fig. 6). The optimal distance of 1.5 mm was determined by varying the separation distance between the tribopositive and tribonegative surfaces and this was maintained throughout the tests (Supplementary Fig. 7). From the voltage output data (Fig. 2c, d), the sharp single peak with highest voltage of 98.5 ± 0.3 V was obtained from the HKUST-1 sample subjected to a 2 h growth time (henceforth denoted as 2 h-HKUST-1). In comparison, the pristine copper plate produced the lowest voltage output of 21.0 ± 0.1 V,

indicating that the growth of a thin dielectric film comprising HKUST-1 on the copper surface enhanced the voltage output. However, the voltage output did not increase proportionally with the growth time. The 3 h growth time resulted in a lower voltage output of 51.9 ± 0.4 V compared to the 1 h growth time. The trend observed in the voltage output was consistent with the trends in current and charge, indicating a correlation between these material parameters. Overall, the highest output was achieved with the 2 h-HKUST-1, which was also reproducible in results determined from three separate batches of fabricated film samples (Supplementary Fig. 8), yielding a peak-to-peak voltage ($V_{pp}$) of 143.2 ± 0.7 V, a maximum charge of 9.8 ± 0.3 nC (charge density ~ 11 µC m$^{-2}$) and a closed-circuit current of 17.8 ± 0.2 µA (current density ~ 20 mA m$^{-2}$), over 4.6–5.8 times higher than the pristine copper plate tested under the same conditions. These results provide a different perspective on the facile attachment of MOFs to substrates, enabling stable and enhanced TENG performance compared to other, more time-consuming MOF growth methods (Supplementary Table 3).

In addition to the standard output performance tests, other parameters were also collected to assess the properties of the TENG based on the 2 h-HKUST-1 TENG. A mechanical stability test was conducted to assess the durability of the 2 h-HKUST-1 TENG. Under the same conditions, the voltage output was continuously monitored over 97,000 impact cycles completed in 13.5 h, Over 13.5 h of continuous operation, the TENG device maintained a stable output of 93.6 ± 1.8 V, with only a decline of 1.9% (Fig. 3a). Magnified view of a few representative cycles is shown in Fig. 3a, Supplementary Fig. 9a and b, highlighting the detailed features of each contact-separation peak. Figure 3b shows an SEM image of the 2 h-HKUST-1 sample after cyclic testing. For comparison, a similar region prior to testing is shown in Supplementary Fig. 9c, while the Kapton counter layer following the same test is given in Supplementary Fig. 9d. Photographs taken before and after the stability test are provided in Supplementary Fig. 9e. A small amount of crystal transfer to the Kapton tape was observed and with microcracking observed in the HKUST-1 crystals, suggesting relatively strong adhesion and mechanical robustness of the electro-synthesised HKUST-1 film. The prepared 2 h-HKUST-1 TENG device was connected in a closed circuit with varying load resistances ranging from 0.1 to 300 MΩ to determine the optimal operating conditions and to evaluate its practical feasibility. With a constant input frequency of 2 Hz and a compression load of 90 ± 10 N, the peak closed-circuit voltage was measured. The power density $P_d$ was calculated using Eq. (5):

$$P_d = \frac{V^2}{R \times A} \qquad (5)$$

where $R$ is the load resistance and $A$ is the effective contact area, which corresponds to the nominal surface area of 9 cm$^2$ employed in this study. As shown in Fig. 3c, the 2 h-HKUST-1 TENG achieved a maximum power density of 771.8 ± 0.3 mW m$^{-2}$ with an increasing voltage trend that ultimately plateaued. Compared with other pure MOF-based TENGs, as summarised in Supplementary Tables 3–5, the 2 h-HKUST-1 TENG, featuring a directly grown MOF thin film that was stable, showed great promise in terms of both voltage output and power density.

Additionally, to evaluate the effectiveness of the 2 h-HKUST-1 TENG as an energy harvester, the alternating current generated by the TENG was collected and stored in capacitors using a full rectifier circuit, as illustrated in Supplementary Fig. 10a. After the induced charges passed through the rectifying circuit, the alternating current was converted to direct current by a network of diodes. This rectified current could then be used to charge capacitors and small commercial electronics. As illustrated in Fig. 3d, the same condition was applied to charge several commercial capacitors, ranging from 0.1 to 47 µF. All capacitors achieved a reasonable charging voltage after a relatively short period of time. In a practical capacitor-charging test, the 0.1 µF capacitor reached 2.5 V within 25 s, demonstrating the feasibility for storing harvested charges and subsequently redistributing them to drive external electronic devices. To verify

this, a 2 V LED was used in conjunction with the 47 µF capacitor. As shown in Supplementary Fig. 10b and Supplementary Movie S1, 48 LEDs could be illuminated with a continuous impact cycle, demonstrating that the 2 h-HKUST-1 TENG could effectively harvest energy to power small electronics such as an array of LEDs. Since TENG is a high-impedance and mostly capacitive source, in the lab scale TENG performance, the peak power ($P = V^2/R$) is always the sweet spot load with an optimal load. In practice, rectifiers, storage capacitors, and control electronics make the effective load time-varying with switch losses, so there is no single fixed value of optimal $R$. In a laboratory setting, we note that the fixed resistor test serves as a useful benchmark to systematically compare the relative performance of a series of materials being developed.

## Mechanism analysis of HKUST-1 TENG

The enhanced triboelectric performance of the 2 h-HKUST-1 TENG can be attributed to the crystallinity and microstructural characteristics of the film grown *via* electrodeposition. As evidenced by the XRD analysis (Supplementary Fig. 4), the 2 h-HKUST-1 sample had a higher crystallinity ratio, particularly highlighted by the high intensity ratio of the (222)/(511) peaks exceeding ~6, indicating completion of crystal growth at this time point. In contrast, the 3 h growth condition resulted in excessive crystal growth (Supplementary Fig. 5v), exhibiting relatively large crystals exceeding ~50 µm in diameter and precipitations observed in solution during the electrodeposition process. In terms of film adhesion, adhesion between HKUST-1 and copper increased with thickness[30]. The similar thickness between the 2 and 3 h growth time samples suggested that their adhesion strength was similar; thus, the reduced output of the 3 h sample was not primarily due to adhesion.

SEM micrographs (Fig. 1d) revealed that the 2 h-HKUST-1 sample had a preferential orientation of the (222) facets. Quantitative XRD peak integration confirmed this observation. The (222) facet showed the largest integrated area, accounting for 35.6% of the total cumulative peak area (Supplementary Fig. 11, Supplementary Table 6). The remaining crystal orientations together contributed 64.5%, each with a smaller share, underscoring the dominance of the (222) facet for the 2 h-HKUST-1 sample. Atomic force microscope (AFM) amplitude profile further revealed dislocation spiral growth consistent with the final stages of crystallisation (Fig. 4a)[33]. These facets naturally aligned facing upward with some differences in height (0.1–0.5 µm) (Supplementary Fig. 12), enhancing contact area with Kapton. Additionally, compared to the (200) facet, the (222) facet demonstrated relatively higher mechanical properties, specifically in hardness (yield strength), $Y_{(111)} \approx 2 \times Y_{(100)}$ and Young's modulus (stiffness), $E_{(111)} \approx 3.6 \times E_{(100)}$[34].

Nanoscale surface potential measurement via Kelvin probe force microscopy (KPFM) (Fig. 4b) further validated the tribopositive nature of HKUST-1, especially on the (222) facets, showing a positive average surface potential of 648.4 ± 7.1 mV. In Supplementary Fig. 13, the local surface potential increased with the size of the (222) facet. For instance, as the facet area rises from 9.6 to 71.9 µm$^2$, the surface potential rose by almost 96%. This observation can be one of the factors contributing to the improved output in the 2 h-HKUST-1 sample, due to the completed crystal growth exhibiting a dominant (222) facet. Together with the increase of the (222) surface area, we reasoned that the higher stiffness of the {111}-oriented crystal planes[34] of HKUST-1 may influence the interfacial contact behaviour. Being relatively harder and stiffer, the (222) facets thereby facilitated an improved contact with the opposing layer that was more pliant. Hence, the triboelectric output was dependent on the contact surface mechanics, and this was a function of the actual contact area of the interface in dynamic motion. Through local-scale nano-FTIR characterisation depicted in Fig. 4c, the second-harmonic optical phase image (O2P) revealed the nanoscale chemical uniformity of the crystal facet. The second-harmonic optical amplitude image (O2A) (Fig. 4d) with a 25-point line scan, each point with ~20 nm spatial resolution, revealed a uniform absorption signal across the (222) facet of HKUST-1 (Fig. 4d–f)[35]. We proposed that nanoscale uniformity facilitates local charge transport across material interfaces, reducing

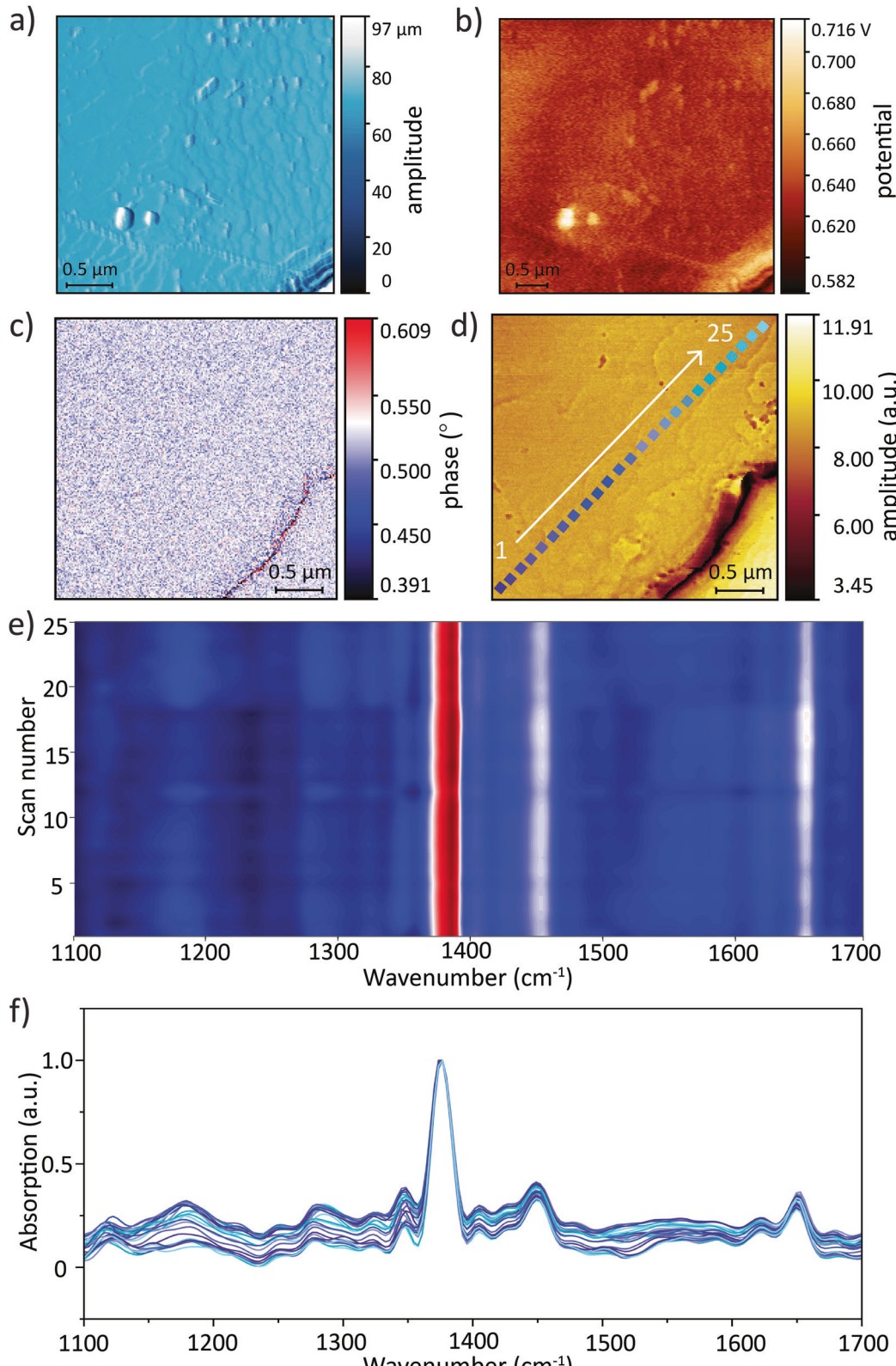

**Fig. 4 | KPFM and nano-FTIR characterisation of 2 h-HKUST-1. a** AFM amplitude image on the single crystal surface. **b** KPFM image of surface potential on the single facet. **c** Second-harmonic optical phase image (O2P). **d** Optical amplitude image (O2A) of a (222) oriented crystal facet denoting the positions of a 25-point line scan. **e** Contour plot of O2P determined from the local positions depicted in (**d**). **f** Nano-FTIR absorption spectra correlated to (**e**) and corresponding to positions marked in (**d**).

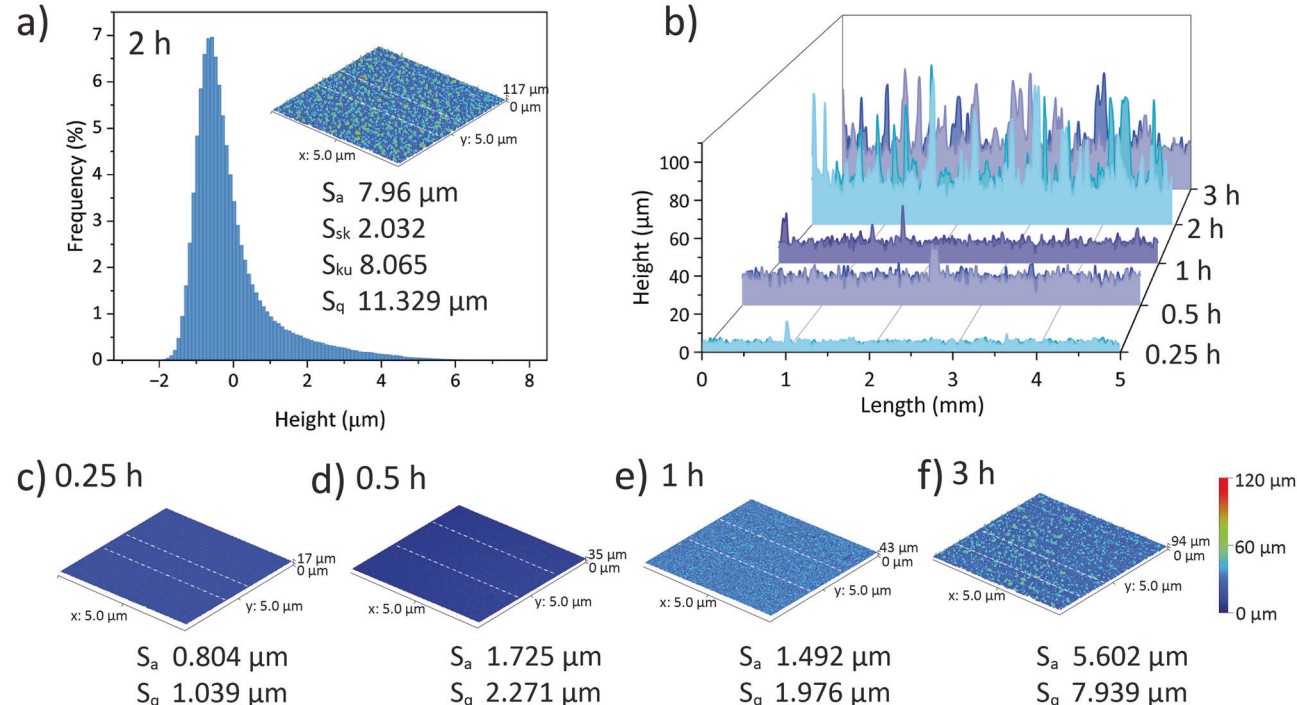

**Fig. 5 | Surface roughness profiles measured by a non-contact Alicona optical profilometer. a** Height distribution and 3D topology of the 2 h-HKUST-1 sample (5 mm × 5 mm) with $S_a$ (average height), $S_{sk}$ (skewness), $S_{ku}$ (kurtosis), and $S_q$ (root-mean-square height). **b** Cross-section profiles of five samples with different growth times, with two selected cross-sections indicated by dashed lines in each 3D height topology (**c–f**). **c–f** 3D surface topography of samples at 0.25, 0.5, 1 and 3 h, respectively.

charge leakage and maintaining stable triboelectric performance under continuous cycles.

Microscale surface roughness also critically impacted triboelectric function. The 2 h-HKUST-1 sample displayed the highest surface roughness parameters, characterised by an average height ($S_a = 7.96\ \mu m$), root-mean-square height ($S_q = 11.329\ \mu m$), high skewness ($S_{sk} = 2.032$), and high kurtosis ($S_{ku} = 8.065$), indicating a surface dominated by numerous moderate height values (Fig. 5a). Compared to shorter reaction times (0.25 h and 0.5 h), where the surface remained relatively uniform with almost even symmetric distributions, the 2 h sample showed a shift toward greater asymmetry and microroughness (Fig. 5b–d, Supplementary Fig. 14). While the 1 h sample already demonstrated increasing skewness and kurtosis (Fig. 5e), the 2 h sample amplified this trend, suggesting a more pronounced growth of micron-sized surface features. In contrast, although the 3 h sample also exhibited substantial microroughness, its slightly reduced skewness and kurtosis suggested a possible stabilisation or saturation of the surface microstructure (Fig. 5f). Additionally, the area factor, $S_{dr}$, was calculated using Eq. (6) to estimate the real contact area based on the measured $S_a$ and $S_q$[36]. From Supplementary Table 7, the 2 h-HKUST-1 sample exhibited the highest $S_{dr}$ of 17.6%, thereby, consistent with the output trend and indicating that the increased additional surface area contributed to the higher output. The additional surface area may cause a larger contact area with Kapton, consistent with the observed output improvement.

$$S_{dr} = \frac{(\text{surface area with texture} - \text{projected surface area})}{\text{projected surface area}} \times 100\% \quad (6)$$

Collectively, the enhanced triboelectric performance of the 2 h-HKUST-1 TENG results from optimal crystallinity, well-oriented crystal facets, nanoscale uniformity and surface microroughness. These structural and morphological characteristics offer critical insights and design guidelines for developing and optimising other MOF-based films as tribopositive materials.

## Effect of humidity on hydrophilic HKUST-1 polycrystalline film

Given the hydrophilic nature of HKUST-1 due to its dimeric cupric tetracarboxylate units, it was worthwhile to study the effect of humidity on the TENG performance. As the $N_2$ purging removed the moisture in the TENG setup, the colour of HKUST-1 immediately switched from light blue to dark blue (Supplementary Fig. 3f), and the TENG output was recorded for different RH levels. Figure 6a and b shows that as RH decreased, the voltage and current output gradually increased for the HKUST-1 samples grown for 0.25, 0.5 and 1 h. However, the output was found to be relatively stable for the samples with growth times of 2 and 3 h. Specifically for the 2 h-HKUST-1 sample, the normalised voltages and associated errors were obtained by dividing each measured value by the maximum voltage (93.5 V). As shown in Supplementary Fig. 15, the voltage did not exhibit a strictly consistent increase with decreasing RH. Minor fluctuations were observed around 40 RH%, which could be attributed to environmental factors such as slight temperature drifts, transient humidity instability, and adsorption–desorption lag effects. The 2 h-HKUST-1 sample was also tested for ~10,000 cycles controlled around 70 RH% to emulate the humid weather, where the output voltage was fluctuating between 89 and 95 V, without a dramatic drop in its performance (Supplementary Fig. 16). After the humidity stability test, FTIR spectra collected before and after humidity exposure (Supplementary Fig. 17) revealed a marked change in the relative peak intensity of the 2923 and 1374 cm$^{-1}$ bands, corresponding to the C–H and C–O vibrational modes. Particularly, the relative increase in the intensity of the 2923 cm$^{-1}$ band, assigned to C–H stretching of the BTC linker, can be attributed to water adsorption within the HKUST-1 pores, which modified the local environment of the organic linker and thus enhanced the vibrational response.

At higher RH levels, water molecules can readily coordinate to the unsaturated $Cu^{2+}$ sites on the paddle wheels[37], forming bonds with the axial Cu–O bonds and filling the pores of the HKUST-1 framework (Fig. 6c). These water molecules can partially neutralise surface charges by providing a conductive medium for surface charge redistribution. Consequently, the

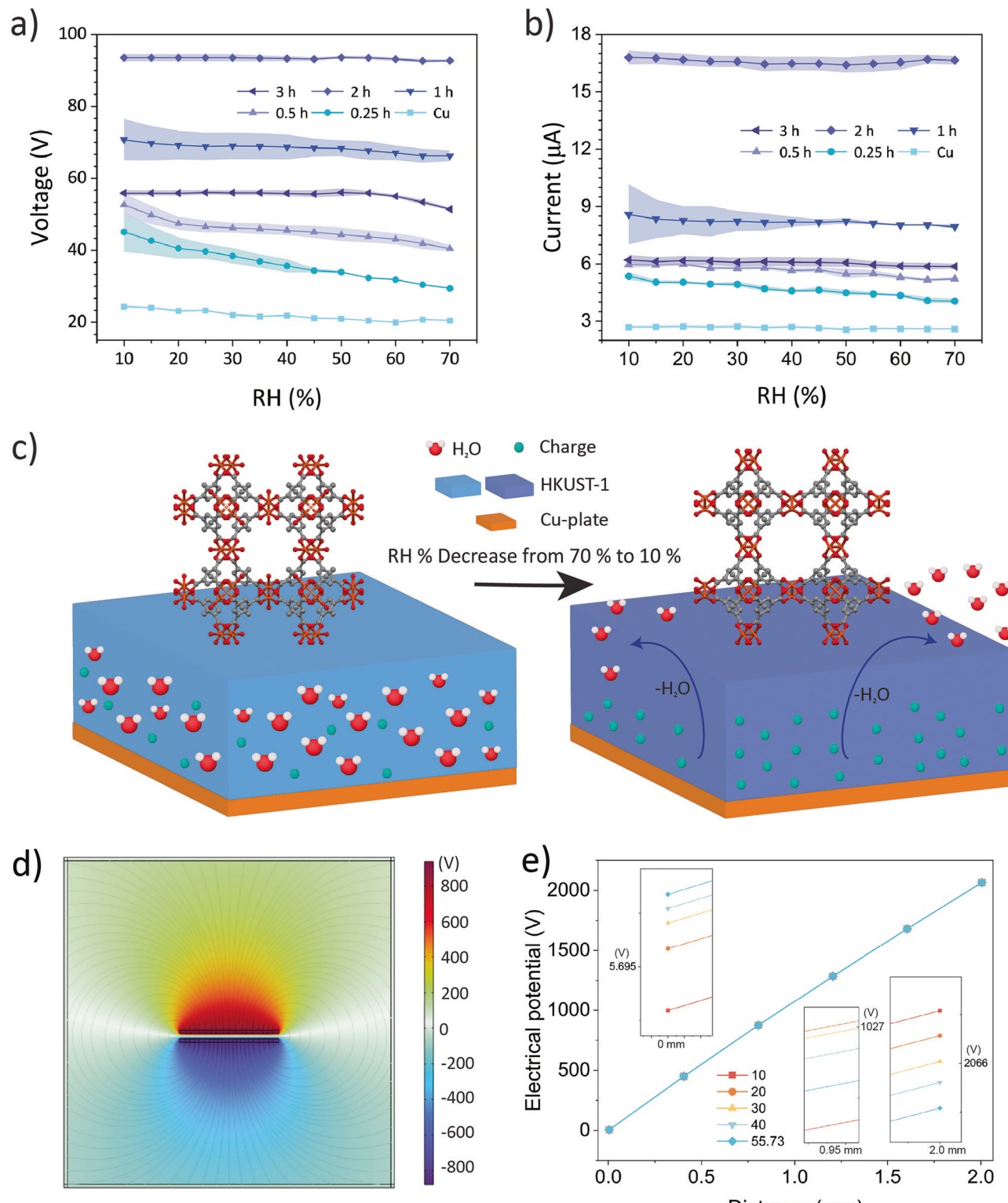

**Fig. 6 | Humidity test across varying relative humidity (RH) levels from 70% to 10%. a** Output voltage and **b** output current as a function of RH%. **c** the proposed mechanism of a hydrophilic MOF surface. **d** Finite-element method (FEM) predictions of electric potentials on the triboelectric material surfaces with a dielectric constant of 55.73 under high RH and 1.5 mm distance. **e** Predicted electric potential as a function of displacement between the pair of electrodes by varying the dielectric constants of the material (10, 20, 30, 40, 55.73).

voltage and current observed at higher RH levels are relatively lower than those recorded at lower RH. As the RH decreases, water molecules desorb from the HKUST-1 structure, reducing the availability of mobile charges and diminishing the charge screening effect[38]. We proposed that such desorption leads to an increase in the active surface area, which enhances the

generation of triboelectric charge during the contact-separation process. At 10 RH%, this results in a greater retention of surface charge, producing higher voltage and current.

Additionally, the {111}-oriented facets of the HKUST-1 crystal are occupied by the hydrophobic aromatic rings of the BTC ligands, which

create small pockets (~0.35 nm in diameter) that are less affected by water molecules, whereas water molecules can preferentially only occupy the larger pores of the (100) facet[27]. This allows charge accumulation to persist on the hydrophobic regions of the surface. The output of HKUST-1 with 2 and 3 h growth time, however, remains nearly constant, which can be explained by the formation of an adsorption-desorption equilibrium attributed to the continuous polycrystalline film. We hypothesise that once this equilibrium is reached, further desorption of water no longer significantly impacts charge transport or surface charge accumulation, thus leading to a saturation effect. This explains why a further reduction in RH did not result in a substantial increase in output. Interestingly, the dense copper plate used as an electrode is also slightly affected by changes in RH. A decrease in RH makes the surfaces of copper and Kapton drier, enhancing charge transfer efficiency when the two materials come into contact and separate. This results in higher output due to a reduction of charge dissipation.

Finite element method (FEM) simulations were conducted to investigate the impact of dielectric constant on the voltage output of HKUST-1 under several conditions for parametric studies: with water (dielectric constant, $\varepsilon = 55.73 \pm 1.87$)[25], and with a wider range of values ($\varepsilon = 10, 20, 30, 40$) used for comparison. The simulations gave us a better insight into how the dielectric constant may influence the voltage output associated with triboelectrification. Figure 6d showed the electric potential of HKUST-1 under a high RH with a dielectric constant of 55.73 (upon water sorption)[25] and at a maximum contact-separation gap of 1.5 mm. Figure 6e showed the electrostatic potential difference ($\phi_{top} - \phi_{bottom}$) as a function of separation distance from 0 to 1.5 mm as denoted in Supplementary Fig. 2, under varying dielectric constants. Notably, when the separation was less than ~0.95 mm, higher humidity (thus greater $\varepsilon_d$) gives higher voltage output, but beyond ~0.95 mm the trend flips, i.e., higher humidity gave a lower voltage. Such a trend reversal with increasing separation distance can be explained by the shift in the dominant mechanism, as given by Eqs. (7) and (8)[3].

$$V_{tri} = \frac{\sigma d}{\varepsilon_0 \varepsilon_d} \tag{7}$$

$$V_{oc} = \frac{\sigma x(t)}{\varepsilon_0} \tag{8}$$

where $V_{tri}$ and $V_{oc}$ are the triboelectric voltage and open circuit voltage, respectively. $\varepsilon_d$ and $d$ correspond to the dielectric constant of the triboelectric material and its thickness; $\varepsilon_0$ is the permittivity of vacuum. When the contact-separation gap size is smaller than 0.95 mm, the dielectric property of the materials has a stronger influence on the voltage output, and hence Eq. (7) applies, where the higher humidity increases $\varepsilon_d$, thereby reducing voltage output. However, the presence of moisture may also enhance the surface charge density, $\sigma$[39]. Even though the higher dielectric constant tends to lower the voltage, the extra surface charges generated under humid conditions can overcome or compensate for that reduction, resulting in a net increase in voltage as the separation gap gradually widens. As the separation distance increases beyond ~0.95 mm, the influence of the dielectric layer becomes negligible, and the voltage is better described by Eq. (8), where the time-varying air gap distance, $x(t)$, dominates the potential drop. In this case, the higher humidity no longer benefits voltage generation, causing a lower output. At 1.5 mm distance, the predicted trend of potential difference aligns well with the experimental measurements, indicating that a lower RH moderately improves the voltage output.

## Discussion and outlook
HKUST-1, with open $Cu^{2+}$ sites and hydrophilic pores, can be electrodeposited and patterned on a solid copper substrate to yield a well-faceted polycrystalline film. At ~70 RH%, modest water uptake sustains interfacial polarisation with low charge leakage, enabling reliable operation of wearables and portable devices. Although sweat or breath can modulate the signal for self-powered humidity sensing, the corrosion of copper should be

considered for practical implementations. Selective adsorption of $CO_2$, $NH_3$, alcohols, and $N_2$ in HKUST-1 could also change its output or the film colour, in which the crystallographic control of the {111}-oriented facets on micro-textured copper may boost efficiency for integration in compact harvesters targeting Internet of Things (IoT). Protection of the device against dust and contaminants could be achieved, for example, by encapsulating the device in a dust-free enclosure or by applying a very thin protective coating compatible with the triboelectric layer. Upscaling of devices beyond 3 cm × 3 cm is feasible by preserving electrochemical uniformity via MOF-based film methodologies.

It is envisaged that limits will arise at a high humidity level of above 80 RH% combined with elevated temperature, because water can slowly hydrolyse Cu–carboxylate bonds[40]. Beyond ~100 °C, the HKUST-1 framework was reported to be more susceptible to structural deformation[41]. In contrast, at low temperatures, structural stiffness increases, but charge mobility falls, and surface moisture can freeze, impeding charge transfer. The performance degradation caused by the high humidity or temperature can be recovered through mild heating or vacuum drying, which removes adsorbed water and restores the coordination structure of the Cu–carboxylate bonds. Performance will therefore deviate from room-temperature behaviour and these factors warrant systematic investigation employing a custom-built TENG setup with precise temperature and humidity controls.

## Conclusions
This study demonstrates a facile electrochemical deposition method to directly grow MOF-based tribopositive materials, specifically the HKUST-1, on a copper substrate. This method produces a stable polycrystalline MOF film with strong adhesion, providing a practical approach for fabricating TENG devices. The 2 h-HKUST-1 film (3 cm × 3 cm) shows good performance, achieving a power density of $771.8 \pm 0.3$ mW m$^{-2}$ with a peak-to-peak output voltage of $143.2 \pm 0.7$ V, a charge output of $9.8 \pm 0.3$ nC, and a closed-circuit current of $17.8 \pm 0.2$ μA, which is 4.6–5.8 times higher than that of the pristine copper substrate. The improved performance is linked to the high crystallinity with well-oriented (222) crystal facets, nanoscale uniformity and surface microroughness. The 2 h-HKUST-1 TENG also maintains stable output with a minimal decrease in ambient and high-humidity environments. Notably, the humidity test suggests that decreasing humidity may lead to an increase in voltage and current output of TENG incorporating HKUST-1 film, indicating that water molecules do not adversely affect performance under ambient conditions pertinent to many practical applications. Together, these results support further exploration of the 2 h-HKUST-1 TENG for compact triboelectric applications and provide guidance for future development of other MOF-based TENGs with polycrystalline films.

## Methods
### Electrochemical deposition of HKUST-1
All chemicals used in this work are commercially available. 1,3,5-benzenetricarboxylic (BTC) acid was purchased from Fisher Scientific, tributyl methylammonium methyl sulfate (MTBS), acetone and ethanol were obtained from Sigma-Aldrich.

Before electrodeposition, the ~0.9 mm-thick, 3 cm × 4 cm Cu plate substrate designated for the anode side was ground to a grit of 4000 to remove surface contamination, followed by polishing to 1 μm using diamond suspensions, as reported in the previous research[30]. The Cu plate for the cathode was prepared to a grit of 2000. The polished substrates were then sonicated for 5 min and submerged in ethanol to remove any residue.

The setup for the electrodeposition device was shown in the supplemental information (Supplementary Fig. 1a). For each reaction, 100 mL of electrolyte was prepared, consisting of 56 mL of ethanol, 44 mL of deionised water, 1 g of BTC ligand, and 2 g of MTBS conduction salt, which were stirred until a clear solution was obtained. The solution was then heated to 55 °C, and the electrodes were immersed in it while a voltage of 2.5 V was applied. The HKUST-1 films grew on the inner surface of the anode over the

**Article**

selected reaction times (0.25, 0.5, 1, 2, and 3 h). After completion of the reaction, the anode with the resulting polycrystalline coating was washed with ethanol to eliminate any unreacted ligands and $Cu^{2+}$ ions.

### Fabrication of the TENG device

The HKUST-1 device was fabricated for testing in the contact-separation mode. The electrodeposited area of HKUST-1, ~3 cm × 3 cm, served as the tribopositive side with a Cu substrate. On the opposite side, an ITO-PET substrate adhered with Kapton tape was used as the negative electrode and tribonegative layer, respectively.

The TENG measurements were conducted under contact-separation mode using a permanent magnetic shaker (Brüel & Kjær LDS V201) coupled to a voltage-amplified arbitrary function generator (GW Instek AFG2105) (Supplementary Fig. 1b). During the standardised test, the amplitude of displacement was maintained at 1.5 mm with a frequency of 2 Hz, and the maximum impact force was maintained at ~90 ± 10 N as shown in Supplementary Fig. 6. The TENG output voltage was monitored by an oscilloscope (Picoscope 5442D) with a 100 MΩ high voltage probe (Rigol RP1300H). The device output current and charge transfer were measured with an electrometer (Keithley 6514).

### Humidity test of HKUST-1 TENG

The TENG device was enclosed within a sealed box, which was fitted with an $N_2$ gas tube to control the internal environment. A humidity sensor was placed inside the TENG setup to monitor real-time humidity levels. The flow speed of the dry nitrogen gas was maintained at 0.2 bar. The experiment was conducted under standardised conditions, maintaining a displacement of 1.5 mm, a frequency of 2 Hz, and a maximum impact force of approximately 90 ± 10 N. During the test, both voltage and current outputs were recorded.

### Characterisation and measurements

The surface morphologies of HKUST-1 were characterised using scanning electron microscopy (SEM: Tescan LYRA & Hitachi TM3030Plus). The crystalline structure of the MOF and nanocomposites was analysed with a Rigaku MiniFlex X-ray diffractometer (XRD), utilising a Cu Kα source (1.541 Å). Fourier-transform infra-red (FTIR) spectra were recorded using a Nicolet iS10 FTIR spectrometer equipped with an attenuated total reflectance (ATR) module. Surface topography and surface potential measurements were performed in an atomic force microscope (AFM) under tapping mode and Kelvin probe force microscopy (KPFM) mode, respectively, on an Oxford Instruments Cypher-ES AFM. Nano-FTIR spectra of the composites were determined from a scattering-type scanning nearfield optical microscope (Neaspec s-SNOM) operating under the tapping mode. The approach was in accordance with the nanoFTIR characterisation of MOFs as described by Möslein et al.[35]. Surface roughness was measured by an optical noncontact 3D profilometer (Alicona Infinite Focus) with ×10 lens.

### Numerical modelling

The electric potential variation was modelled using the finite element method (FEM) in the COMSOL Multiphysics software. An electrostatics analysis approach was applied, and the device geometry was designed to replicate the experimental setup, featuring electrodes and composite material with a nominal contact area of 3 cm × 3 cm and a maximum contact-separation gap of 1.5 mm between the electrodes (Supplementary Fig. 2). ITO-PET was chosen as the tribonegative electrode, while the copper plate served as the tribopositive electrode. The surface charge densities of the HKUST-1 films and Kapton were defined based on experimental data, and the dielectric constant values were obtained from the literature[25]. As boundary conditions, the electrical potential for the top and bottom electrodes was fixed at 0 V (i.e., grounded) as shown in Supplementary Fig. 2. Steady-state electrostatics calculations were conducted by varying the displacement (0–1.5 mm) between the electrodes to predict the resulting electric potentials generated across the triboelectric device. The model input parameters are summarised in Supplementary Table 1.

## Data availability

Numerical source data for graphs are available within the data set of the supplementary information section (Supplementary Data 1 and 2).

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

## Acknowledgements

This work was supported by the EPSRC Frontier Research Grant (TEGMOF grant EP/Z534146/1). We thank Dr. Jinke Chang for scientific discussion; Dr. Cyril Besnard for help on SEM; Dr. Jiahao Ye for help with TENG device assembly and Tianhuai Xu for help on the Alicona optical profilometer.

## Author contributions

C.Z.J. conceived the project with input from J.C.T., prepared the crystals, performed data analysis, and wrote the first draft of the manuscript; J.C.T. supervised C.Z.J., revised the draft manuscript, discussed the results and contributed to the final version of the manuscript with C.Z.J.

## Competing interests

The authors declare no competing interests.
