## [Transparent Peer Review file · Communications Chemistry]

Robust triboelectric energy harvesters engineered from electrochemically deposited films of HKUST-1 polycrystals

Corresponding Author: Professor Jin-Chong Tan

Version 0:

Reviewer comments:

Reviewer #1

(Remarks to the Author)

In this paper the authors have designed a prototype of the triboelectric energy harvester based on the MOF HKUST-1 polycrystal film. This new triboelectric nanogenerator (TENG) is found to exhibit good stability and triboelectric performance even under a high humidity. Overall, this paper is interesting and well written. I recommend its publication in Communications Chemistry. However, there some issues needed to be clarified before its publication.

1. The explanation to why the 2-h HKUST-1 film exhibits the best performance is not that convincing. According to Figures 1a and S4, the XRD patterns of 2-h and 3-h HKUST-1 films exhibit a very small difference, while the difference of their triboelectric performances is very large. I am wondering that is the thickness of HKUST-1 films another important parameter significantly affecting the triboelectric performances. Have authors measured the thicknesses or imaged the cross section of HKUST-1 films with different growth times?
2. In Figure 4b, the authors claimed that the HKUST-1 film surface exhibits a significant positive potential. By carefully comparing Figure 4b to Figure 4a, the KPFM profile (Figure 4b) is extremely similar the height profile (Figure 4a). The z (out-of-plane position) dependence of potential is usually regarded as a possible source of error in the conventional KPFM measurements (Adv. Funct. Mater. 2012, 22, 652–660). So, did the authors exclude this effect from the KPFM measurements?
3. For a clear comparison, authors can provide the SEM images of the same region of HKUST-1 film and Kapton before and after extended 2-day contact-separation tests.
4. Have the authors verified the stability of the present TENG under the environment of high humidity?
5. The authors claimed that “the presence of moisture can also enhance the surface charge density” of HKUST-1. Do there exist any evidences to support this statement?
6. Explanation of each subgraph was missing from Figure S11.

Reviewer #2

(Remarks to the Author)

The manuscript “ Robust triboelectric energy harvesters engineered from electrochemically deposited films of HKUST-1 polycrystals” is not having any potential novelty for the publications in Nature Communications Chemistry. I strongly recommend a rejection

1. Novelty claimed by author (direct growth of MOFs on conducting substrates or direct growth of MOFs and utilized in TENG design) is well reported in the literature.
(ex: <https://pubs.rsc.org/en/content/articlehtml/2023/ta/d3ta03231k>,
<https://pubs.rsc.org/en/content/articlelanding/2018/nj/c7nj03171h>,
<https://pubs.acs.org/doi/full/10.1021/acssuschemeng.3c05198>,
<https://advanced.onlinelibrary.wiley.com/doi/full/10.1002/aenm.201803581>)
2. There are many TENG reports on HKUST (ex: <https://advanced.onlinelibrary.wiley.com/doi/full/10.1002/adfm.201807655>)
3. The output power of the TENG reported is small compared to the existing MOF-based TENGs.
4. There is no application part of the work. Most TENG reported in the literature are good and novel applications.
5. In the stability graph, data fabrication is there. Same data graphs pasted chk the profiles of the bottom. The spike pattern is

identical in both the upper and lower regions.

Reviewer #3

(Remarks to the Author)

The manuscript presents a direct electrochemical deposition strategy for fabricating robust HKUST-1 films on copper substrates for triboelectric nanogenerators (TENGs). The approach is interesting, the dataset is extensive, and the work demonstrates notable output performance and durability compared to previously reported MOF-based TENGs. However, several aspects require clarification, deeper analysis, and improved contextualization before the work can be considered for publication in a Nature-family journal. Below, I outline the key concerns and specific queries for the authors. The strength of the manuscript lies in its novel direct-growth approach, comprehensive characterization, and environmental testing. Nonetheless, there are inconsistencies in the interpretation of humidity effects, limited benchmarking against the state-of-the-art, and insufficient quantitative correlation between structure and device performance. Addressing the points below will significantly strengthen the impact and clarity of the work.

1. Please reconcile the apparent contradiction between the conclusions (p. 20, lines 473–475) stating humidity increases output and Figure 6a–b, which generally shows output decreasing with RH for most samples.
 2. Provide quantitative data (e.g., XRD peak integration or image analysis) on the fraction of (222) facets vs. other orientations in the optimal 2 h sample.
 3. Estimate the real contact area increase due to the measured surface roughness (S_a , S_q) and relate it quantitatively to observed output enhancement.
 4. Report performance variation over multiple fabrication batches for the 2 h growth condition to demonstrate reproducibility.
 5. Quantify the adhesion strength of the electrodeposited HKUST-1 film (e.g., via scratch testing, tape test, or mechanical pull-off).
 6. Provide data or discussion on chemical/mechanical degradation after extended cycling in high-RH conditions.
 7. Have temperature-dependent performance tests been conducted? If not, discuss expected performance at elevated and reduced temperatures.
 8. Present data or discussion on scaling the deposition beyond $3 \times 3 \text{ cm}^2$, including any uniformity challenges.
 9. In the KPFM analysis, quantify the correlation between facet size and local surface potential variation.
 10. Provide statistical metrics from the nano-FTIR maps to support the claim of nanoscale chemical uniformity.
 11. In FEM simulations, clarify how the surface charge density values were determined and whether they vary with RH in the model.
 12. Include a main-text figure comparing your device performance (voltage, current, power density) to leading MOF-based and non-MOF TENGs.
 13. For capacitor charging experiments, calculate the mechanical-to-electrical energy conversion efficiency.
 14. Discuss optimal load matching for maximum power transfer and how this compares with real application scenarios.
 15. Have XPS or related analyses been performed before and after high-RH exposure to check for chemical changes?
 16. Explain whether the performance is strongly dependent on using copper as the substrate; could other conductive substrates be used?
 17. Provide direct or literature-based quantitative values for Young's modulus and hardness of (222) vs. other facets to support mechanical stability claims.
 18. In long-term cycling tests, provide statistical analysis of any subtle decay trends over 97,000+ cycles.
 19. Discuss potential sensitivity to dust/organic contaminants on the HKUST-1 surface and how this might affect real-world operation.
 20. Expand the discussion on where such MOF-based TENGs could be uniquely impactful compared to other tribopositive materials (e.g., in marine sensors, wearable electronics).
- I recommend major revision. The work is promising and could make a valuable contribution to the TENG and MOF materials field, but addressing these queries with additional data, analysis, and clarifications is essential to meet the rigor and impact expectations.

Version 1:

Reviewer comments:

Reviewer #1

(Remarks to the Author)

The authors have well answered the questions. I thus recommend the publication of this paper.

Reviewer #2

(Remarks to the Author)

The revised manuscript, "Robust triboelectric energy harvesters engineered from electrochemically deposited films of HKUST-1 polycrystals," has partially addressed the comments. Still the novelty of the manuscript is not up to the mark to publish in such prestigious journal like Nature Communications Chemistry. A few trials can achieve the direct growth of MOFs on conducting substrates by electrodeposition, and the manuscript does not discuss what the thumb rules/guidelines are for electrodeposition of MOFs on conducting substrates in general sense. Further, the application part is so weak, as per

the journal's broad audience, it should be a new potential application. Based on TENG performance, it is not particularly high-performance. Based on the above, I strongly recommend a rejection. However, I am giving clearer comments to improve the manuscript.

1. There is no reason why the author chose Kapton as the opposite layer. As the author expected a positive nature of the HKUST, in that they should have chosen more tribonegative films like (FEP, PDMS, Silicone) as an opposite to get better TENG performance.
2. Authors should provide real-time testing videos for TENG testing under different humidity levels and powering LEDs.
3. The author should provide photographs of the HKUST film before and after stability of 10 K cycles.
4. Author should make a comparison table of the performance of direct growth MOF-based TENGs in the main manuscript (ex: ZIF-67, ZIF-8, MOF-303 etc)
5. The author should provide the test set-up used for TENG testing (photo/video), as the authors used a different set-up compared to the typically used linear motors.
6. Authors should provide force, area, frequency dependent studies to have complete picture of HKUST based TENGs.
7. Even though the authors used a 100X probe, the oscilloscope still gets loaded due to impedance mismatch, which leads to an error in measuring the output voltage. Justify.
8. How did the authors measure current with an electrometer? Current measurements with SR 570 give a more accurate value of currents.
9. Authors should provide charge transfer measurements and the efficiency of the present TENG.
10. Few cycles from the stability can be shown in Zoom.
11. Authors should provide work function data of both the frictional layer (HKUST and PET) to get clarity on charge transfer and the triboelectric nature of the HKUST.

Reviewer #3

(Remarks to the Author)

The authors have made substantial and thoughtful revisions in response to previous reviews. The updated data, added figures (e.g., Figures S8, S11, S13, S15–S16), and clarifications in methodology significantly strengthen the manuscript. The work presents a clear demonstration of a directly electrodeposited HKUST-1 film as a tribopositive layer with stable TENG performance under high humidity, which has both scientific merit and practical relevance.

The paper is now close to publication, pending minor clarifications and small improvements as listed below.

1. The discussion (page 19) explains the interplay between dielectric constant, water adsorption, and charge screening at varying RH.
→ Could the authors include a concise table or plot summarizing normalized voltage versus humidity for the optimal 2 h sample to facilitate direct comparison across RH values (10–70 %)?
2. Figure 3b and Figure S9a–b show pre-/post-cycling SEM images.
→ Please specify whether the observed microcracks (page 11) affect triboelectric output reproducibility over multiple batches, or if the output variance (± 1.6 V) in Figure S8 already accounts for such morphological changes.
3. Page 13–14 highlights the dominance of the (222) facet and references prior mechanical data ($E(111) \approx 3.6 \times E(100)$).
→ Would the authors consider adding a short sentence quantifying how this stiffness difference might translate to expected contact area or charge density variation (qualitative scaling estimate)?
4. The manuscript discusses hydrolysis of Cu–carboxylate bonds above 80 % RH and 100 °C (page 20).
→ Could the authors comment briefly on the expected recovery or regeneration strategy (e.g., mild heating or vacuum drying) for restoring performance after extended humid exposure?
5. Several supplemental tables (S3–S7) and figures are cited.
→ Please ensure all new supplemental materials are sequentially numbered and cross-referenced consistently in both the main text and SI, as minor mismatches were noted between figure numbers in text (e.g., S11/S13 renaming).

Version 2:

Reviewer comments:

Reviewer #2

(Remarks to the Author)

The revision is not satisfactory. Most reviewer comments have been omitted or addressed only with brief logical explanations, without performing the necessary additional experiments. For example, the inclusion of a comparison table is essential—since the authors claim a TENG based on direct growth of MOFs, such comparative data is valuable, especially given the limited literature in this area. Several other important comments also remain inadequately addressed.

Reviewer #3

(Remarks to the Author)

The revised manuscript is now acceptable.

Department of Engineering Science
University of Oxford
Parks Road
Oxford OX1 3PJ
United Kingdom

Prof. Jin-Chong Tan
Professor of Nanoscale Engineering
Fellow of Balliol College, Oxford.
Telephone: +44 (0)1865 273925
e-mail: jin-chong.tan@eng.ox.ac.uk
URL: <https://eng.ox.ac.uk/mmclab>

18 October 2025

Dear Editor,

Manuscript ID: COMMSCHEM-25-0632A.R1

Many thanks for providing the comments from the referees, we greatly appreciate the feedback and suggestions to enhance the quality of the manuscript. We have conducted new experiments and performed further analysis which substantiated the claims made in the manuscript. Point-by-point response to the reviewers' comments are presented below. Changes made to the manuscript are **highlighted** in the revised document and amended Supplemental Information (SI).

Reviewer #1 (Remarks to the Author):

In this paper the authors have designed a prototype of the triboelectric energy harvester based on the MOF HKUST-1 polycrystal film. This new triboelectric nanogenerator (TENG) is found to exhibit good stability and triboelectric performance even under a high humidity. Overall, this paper is interesting and well written. I recommend its publication in Communications Chemistry. However, there some issues needed to be clarified before its publication.

1. The explanation to why the 2-h HKUST-1 film exhibits the best performance is not that convincing. According to Figures 1a and S4, the XRD patterns of 2-h and 3-h HKUST-1 films exhibit a very small difference, while the difference of their triboelectric performances is very large. I am wondering that is the thickness of HKUST-1 films another important parameter significantly affecting the triboelectric performances. Have authors measured the thicknesses or imaged the cross section of HKUST-1 films with different growth times?

Response:

Thank you for highlighting the thickness effect. We measured the total cross-sectional thickness with the copper layer in place, as the Cu substrate is part of the device stack. To minimize variability, all copper substrates were polished to a uniform thickness of ~0.9 mm. Under these controlled conditions, the total thickness (Cu + HKUST-1) increases monotonically with growth time (see Table below). The average thickness was taken from four different locations of each HKUST-1 polycrystalline sample. At 0.25 h the crystals did not

fully cover the immersed copper surface, so a representative film thickness could not be determined.

Growth time / h	Thickness / mm
0	0.911 ± 0.001
0.5	0.925 ± 0.004
1	0.933 ± 0.001
2	0.951 ± 0.005
3	0.976 ± 0.009

It can be seen from the table that, the film thickness scales with the growth time; however, within the measured range it is not the dominant factor for the output performance. The variability for the 3 h HKUST-1 sample mainly reflects height differences across the surface, not a systematic change in thickness. Large-crystal pile-up in the 3 h film (Fig. S5v) leads to early material transfer onto Kapton, modifying the interface and reducing output, even though the thickness is similar to the 2 h film.

The table was added as Table S2 in the SI, with an amendment on page 7: **the average thickness was calculated from four different locations on the electrode grown with HKUST-1 (Table S2), excluding the sample with 0.25 h growth time due to the incomplete crystal coverage.**

2. In Figure 4b, the authors claimed that the HKUST-1 film surface exhibits a significant positive potential. By carefully comparing Figure 4b to Figure 4a, the KPFM profile (Figure 4b) is extremely similar the height profile (Figure 4a). The z (out-of-plane position) dependence of potential is usually regarded as a possible source of error in the conventional KPFM measurements (Adv. Funct. Mater. 2012, 22, 652–660). So, did the authors exclude this effect from the KPFM measurements?

Response:

Thank you for this suggestion. We have considered this effect during the KPFM measurements. To reduce such effects, all KPFM data were acquired in dual-pass mode with a constant and small oscillation amplitude and a fixed lift height, ensuring stable electrostatic force detection. Figure 4a of the amplitude profile shows a small height variation on the facet, thereby the z dependence is not an issue. Therefore, the surface potential determined on the facet such as Figure 4b is reliable.

3. For a clear comparison, authors can provide the SEM images of the same region of HKUST-1 film and Kapton before and after extended 2-day contact-separation tests.

Response:

We agree with the reviewer's suggestion. Representative SEM images were added as Figure 3b (after) and Figure S9a (before). Since SEM operates at the micro- or even nanoscale, it is difficult to relocate the exact same position. We tried our best to find the nearest position on the sample.

On page 11 of the amended manuscript, we have added the following sentences:

Figure 3b shows an SEM image of the 2 h-HKUST-1 after cyclic testing. For comparison, a similar region before testing was shown in Figure S9a, and the Kapton counter layer after the same test is depicted in Figure S9b.

4. Have the authors verified the stability of the present TENG under the environment of high humidity?

Response:

As suggested by the reviewer, we have added stability results under high humidity (~70% RH) to the SI and expanded the explanation in the main text.

Please see the revision on page 17:

The 2 h-HKUST-1 sample was also tested for ~10,000 cycles controlled around 70 RH% to emulate the humid weather, where the output voltage was fluctuating between 89 V and 95 V, without a dramatic drop in its performance (Figure S15).

Figure S15. Stability test of the 2 h-HKUST-1 device with ~10,000 cycles at high humidity, ranging from 70 RH% to 71.8 RH%.

5. The authors claimed that “the presence of moisture can also enhance the surface charge density” of HKUST-1. Do there exist any evidences to support this statement?

Response:

Thank you for pointing this out. From literature, it was reported that when the HKUST-1 coordinates with water, it forms proton-conducting networks, which enhance alternating-current conductivity and interfacial polarization (J. Am. Chem. Soc. 2012, 134, 51–54). In a TENG, the increase in interfacial permittivity and dipole orientation is expected to raise the surface charge density, at moderate RH. This effect warrants further investigation in the context of TENG.

The reference above was cited on page 18 as reference 39.

6. Explanation of each subgraph was missing from Figure S11.

Response:

Thank you for pointing this out. We have improved the captions and quantitatively analysed with added explanations for Figure S13 (previously Figure S11).

Please see the revision on page S19 and page 13 with added sentence:

In Figure S13, the local surface potential increased with the size of the (222) facet. For instance, as the facet area rises from $9.6 \mu\text{m}^2$ to $71.9 \mu\text{m}^2$, the surface potential rose by 95.9%. This observation can be one of the factors contributing the improved output in the 2 h-HKUST-1 sample, due to the completed crystal growth exhibiting a dominant (222) facet.

Figure S13. KPFM surface potential images of the (222) facets. HKUST-1 crystals chosen after electrochemical deposition for a) 2 h, b) 0.5 h, and c) 1 h. Individual facets were masked (white) to calculate the average surface potentials on each facet.

Reviewer #2 (Remarks to the Author):

The manuscript “ Robust triboelectric energy harvesters engineered from electrochemically deposited films of HKUST-1 polycrystals” is not having any potential novelty for the publications in Nature Communications Chemistry. I strongly recommend a rejection

1. Novelty claimed by author (direct growth of MOFs on conducting substrates or direct growth of MOFs and utilized in TENG design) is well reported in the literature.

(ex: <https://pubs.rsc.org/en/content/articlehtml/2023/ta/d3ta03231k>, <https://pubs.rsc.org/en/co>

[ntent/articlelanding/2018/nj/c7nj03171h](https://pubs.acs.org/doi/full/10.1021/acssuschemeng.3c05198), <https://pubs.acs.org/doi/full/10.1021/acssuschemeng.3c05198>, <https://advanced.onlinelibrary.wiley.com/doi/full/10.1002/aenm.201803581>)

Response:

We appreciate the reviewer's feedback. We are aware of the literature mentioned by the reviewer. We acknowledge that MOF TENG studies have been widely investigated. However, the novelty of our paper was using electrodeposition to fabricate a stable device and in demonstrating its stability under high cycle and high humidity conditions. As summarised in Tables S3, S4, and S5, we found no prior MOF-based TENG that was fabricated by electrodeposition. We also tested *ex situ* adhesion and hydrothermal growth routes, which proved unstable: crystals detached and transferred to the counter layer, hindering its use as TENG device.

2. There are many TENG reports on HKUST
(ex: <https://advanced.onlinelibrary.wiley.com/doi/full/10.1002/adfm.201807655>)

Response:

Thank you for the comments. We recognise that the chemical, physical, and mechanical properties of HKUST-1 have been widely explored. However, our study demonstrates pristine crystalline HKUST-1 thin films as the *tribopositive* layer and reveals their basic operating mechanism under ambient and high-humidity conditions. These are novel aspects of our study. On the contrary, the paper cited by the reviewer did not address this case; it blended HKUST-1 with PDMS as a *tribonegative* layer, which is a different device concept.

3. The output power of the TENG reported is small compared to the existing MOF-based TENGs.

Response:

Thank you for pointing this out. We acknowledge that the output power is not the highest among reported MOF TENGs. However, few studies evaluate performance at high humidity; our HKUST-1 device can maintain a stable output at ~70% RH, which sets it apart and makes it suitable for deployment in humid environments, including coastal or marine sensing.

4. There is no application part of the work. Most TENG reported in the literature are good and novel applications.

Response:

Our focus here is on mechanism, combining local nanoscale characterization (for example, nano-FTIR for nearfield measurements) with humidity-dependent measurements to propose a working mechanism, which acts as a proof of concept for sensing in more extreme conditions. Therefore other potential applications are outside the present scope and will be pursued in future.

Here we extend the discussion in more potential applications based on our results. Practical uses include coastal and marine sensor (humidity, salinity spray, wave or vibration harvesting), and wearable patches where breath or sweat modulates the signal for self-powered respiration or perspiration tracking. In air-quality monitoring, HKUST-1 retains adsorption toward CO₂, NH₃ and low-alcohol vapours at elevated humidity, so HKUST-1 TENG output shifts can be used for leak or exposure alerts, with hydration-induced colour change providing a passive indicator.

5. In the stability graph, data fabrication is there. Same data graphs pasted chk the profiles of the bottom. The spike pattern is identical in both the upper and lower regions.

Response:

Thanks for spotting this mistake. We apologise for the confusion caused due to the incorrect figure. The graph has been updated in the revised manuscript on page 12, with the correct data as shown in Figure 3a (copied below).

Reviewer #3 (Remarks to the Author):

The manuscript presents a direct electrochemical deposition strategy for fabricating robust HKUST-1 films on copper substrates for triboelectric nanogenerators (TENGs). The approach is interesting, the dataset is extensive, and the work demonstrates notable output performance and durability compared to previously reported MOF-based TENGs. However, several aspects require clarification, deeper analysis, and improved contextualization before the work can be considered for publication in a Nature-family journal. Below, I outline the key concerns and specific queries for the authors. The strength of the manuscript lies in its novel direct-growth approach, comprehensive characterization, and environmental testing. Nonetheless, there are inconsistencies in the interpretation of humidity effects, limited benchmarking against the state-of-the-art, and insufficient quantitative correlation between structure and device performance. Addressing the points below will significantly strengthen the impact and clarity of the work.

1. Please reconcile the apparent contradiction between the conclusions (p. 20, lines 473–475) stating humidity increases output and Figure 6a–b, which generally shows output decreasing with RH for most samples.

Response:

We thank the reviewer for pointing this out, the general humidity trend is updated in conclusion. As shown in Figures 6a and 6b, lower RH yields higher output, most clearly for the 0.25 h and 0.5 h samples. For the 2 h sample, the output is nearly constant with only minor fluctuations around 92 to 94 V.

The amended sentence was in page 20: The results show that decreasing humidity led to an increase in voltage and current output, indicating that water molecules do not adversely affect performance.

2. Provide quantitative data (e.g., XRD peak integration or image analysis) on the fraction of (222) facets vs. other orientations in the optimal 2 h sample.

Response:

We have added the suggested material to the SI (Figure S11), including the data processing procedure below.

Additional explanation has been added and highlighted on page 13:

... Quantitative XRD peak integration confirmed this observation. The (222) facet showed the largest integrated area, accounting for 35.6% of the total cumulative peak area (Figure S11, Table S6). The remaining crystal orientations together contributed 64.5%, each with a smaller share, underscoring the dominance of the (222) facet for the 2 h-HKUST-1 sample.

Figure S11. Integrated peak areas highlighted in different colours for distinctive facets of HKUST-1.

Table S6. Table S6. Integrated areas, Bragg peak heights after baseline correction, and relative intensity of the 10 strongest reflexions in Figure S11, as calculated by (peak area / total peak area) \times 100%.

Miller index (hkl)	2 θ / °	Area Integration / (au) ²	Max. Height / au	Relative intensity (%)
(222)	12.34	4406.8	28692.3	35.6
(773)	37.11	1870.1	8596.2	15.1
Peak from copper substrate	11.94	1624.2	912.9	13.1
(220)	10.20	832.2	5570.9	6.7
(511)	18.20	634.6	3735.0	5.1
(400)	14.16	519.5	3147.3	4.2
(555)	30.14	419.4	1467.3	3.4
(440)	19.78	362.9	2295.7	2.9
(553)	26.72	282.3	1714.1	2.3
(331)	15.37	236.7	1434.7	1.9
(200)	7.42	228.2	1575.7	1.8

Peak integration was carried out using the Fit Peaks module in OriginPro 2025. The PXRD pattern of the 2 h HKUST-1 sample was baseline-treated with the constant baseline mode (minimum $y = 852$). The 10 strongest diffraction peaks were selected, including a minor hump from the copper substrate (Figure S11). Integrated areas and relative intensities are listed in Table S6.

3. Estimate the real contact area increase due to the measured surface roughness (S_a , S_q) and relate it quantitatively to observed output enhancement.

Response:

Thanks for the suggestion. Direct estimation of the true contact area is challenging. Instead, we considered the area factor, S_{dr} , which is the ratio between the area of the “real” developed surface and the area of the “projected” surface (Stout, K. J. (2000). Development of methods for the characterisation of roughness in three dimensions.)

$$S_{dr} = \frac{(\text{surface area with texture} - \text{projected surface area})}{\text{projected surface area}} \times 100\% \quad (6)$$

The revision can be found on page 14 and Table S7:

Additionally, the area factor, S_{dr} , was calculated using eq. (6) to estimate the real contact area based on the measured S_a and S_q .³⁶ From Table S7, the 2 h-HKUST-1 sample

exhibited the highest S_{dr} of 17.64%, thereby consistent with the output trend and indicating that the increased additional surface area contributed to the higher output. The additional surface area may cause larger contact area with Kapton, consistent with the observed output improvement.

Growth time / h	Area with texture / μm^2	S_{dr} / %
0.25	25.2	0.6
0.5	25.7	2.8
1	25.5	2.0
2	29.4	17.6
3	28.0	12.1

4. Report performance variation over multiple fabrication batches for the 2 h growth condition to demonstrate reproducibility.

Response:

Thanks for the suggestion. We prepared three independent 2 h-HKUST-1 plates and fabricated a TENG device from each one. Each device was tested at least three times: immediately after fabrication, after a few days, and after several months, using either the same or a different Kapton counter dielectric material. In addition to the highest voltage reported in the main text, Figure S8 compiles measurements taken a few days after fabrication of each device.

The revision can be found on Page 9 and Figure S8:

Overall, the highest output was achieved with the 2 h-HKUST-1, which was also reproducible in results determined from three separate batches of fabricated film samples (Figure S8), ...

Figure S8. Reproducibility shown from three batches of samples, with an average voltage of 95.4 ± 1.6 V.

5. Quantify the adhesion strength of the electrodeposited HKUST-1 film (e.g., via scratch testing, tape test, or mechanical pull-off).

Response:

It is true that adhesion may affect triboelectric performance. We have established good understanding of the adhesion strength of polycrystalline MOF films based on HKUST-1. Specifically, our previous work (Cryst. Growth Des. 2015, 15, 1991–1999) concluded that the film-to-substrate adhesion strength is controlled by the film thickness.

Based on the newly added thickness data in Table S2, the revision can be found on page 13:

In terms of film adhesion, adhesion between HKUST-1 and copper increases with thickness.³⁰ The similar thickness between the 2 h and 3 h growth time samples suggested that their adhesion strength is similar, thus, the reduced output of the 3 h sample was not primarily due to adhesion.

6. Provide data or discussion on chemical/mechanical degradation after extended cycling in high-RH conditions.

Response:

Thank you for the suggestion. According to the literature, after extended cycling at high RH and ambient temperature the characteristic XRD reflections of HKUST-1 remain visible, indicating that the crystalline framework is largely preserved. This is consistent with literature showing substantial water uptake of $\sim 33.3 \text{ mmol g}^{-1}$ at up to 90% RH (Chem. Eng. J. 2015, 281, 669–677). Nevertheless, strong coordination of water to the coordinatively unsaturated Cu(II) sites can promote hydrolysis of Cu carboxylate bonds. Under sustained humidity, water molecules near the metal centres may gradually displace BTC ligands, weakening or breaking the metal ligand coordination, an instability also observed in other MOFs such as MOF-5. Mechanically, repeated cycling in humid conditions can cause swelling and shrinkage, which lead to microcracking, interfacial decohesion, and debris formation. These changes modify surface roughness and real contact area that subsequently reduce the electrical output.

The revision is implemented on page 20: ... **water can slowly hydrolyse Cu–carboxylate bonds.**⁴⁰

7. Have temperature-dependent performance tests been conducted? If not, discuss expected performance at elevated and reduced temperatures.

Response:

Thank you for the excellent suggestion. Temperature-dependent testing is challenging with our current setup because of equipment limitations and the safety risks of heating or cooling the assembled TENG setup to non-ambient temperatures. We would need a custom fixture that can independently control the temperature of the two tribolayers, which is outside the present scope but is planned for future development.

From a mechanistic standpoint, TENG output depends on surface morphology, mechanical properties such as adhesion and stiffness, dielectric constant, and environmental conditions. At temperatures above ~ 100 °C, thermal fluctuations can distort the HKUST-1 crystals (Bing, J.Mater.Chem.). The Young's modulus and hardness are expected to decrease with temperature, so the film deforms more during contact–separation test, which should lower the output. At temperatures below 100 °C, literature indicates only a small change in dielectric constant (about 0.2) (Adv. Mater. Interfaces 2020, 7, 2000408). At very low temperatures (for example, around -175 °C), bulk, Young's and shear moduli tend to increase, but charge mobility can decrease, and moisture may condense or freeze on the surface, impeding charge transfer. Taken together, these effects suggest that the output would be reduced and more variable away from room temperature.

The revision is given on page 20.

8. Present data or discussion on scaling the deposition beyond 3×3 cm², including any uniformity challenges.

Response:

We appreciate the reviewer's suggestion. Our deposition method follows Buchan *et al.*'s work (Cryst. Growth Des. 2015, 15, 1991–1999), which deposited HKUST-1 on a 2 cm \times 1 cm copper plate. In this study we scaled the area to 3 cm \times 3 cm.

Scaling beyond 3×3 cm² should be feasible provided several conditions are met. For example, by increasing bath volume proportionally to maintain concentrations, deliver stable and homogeneous current density for ≥ 1 h, enlarge the counter electrode, control temperature, and use current-distribution aids.

9. In the KPFM analysis, quantify the correlation between facet size and local surface potential variation.

Response:

Thank you for the suggestion. The selected KPFM images were analysed in Gwyddion. Individual facets were masked to estimate the mean surface potentials as shown in Figure S13 of the revised SI.

10. Provide statistical metrics from the nano-FTIR maps to support the claim of nanoscale chemical uniformity.

Response:

We appreciate the reviewer's suggestions. We have added an O2P phase image in Figure 4(c), which demonstrates the uniformity of the nano-FTIR signal.

Additional explanation was added on Page 13 of the revised manuscript: **Through local scale nano-FTIR characterisation depicted in Figure 4c, the optical phase image (O2P)**

revealed the uniformity of the facet. A 25-point line scan with ~ 20 nm spatial resolution confirmed uniform local signals across the (222) facets of HKUST 1 (Figure 4d-4f).³¹

Figure 4. KPFM and nano-FTIR characterisation of 2 h-HKUST-1. (a) AFM amplitude image on the single crystal surface, and (b) KPFM image of surface potential on the single facet. (c) Optical phase image (O2P) shows the uniformity of the nano-FTIR signal; (d) Optical amplitude image (O2A) shows the layered crystal surface with 25 points selected for the line scan; (e) Line scan of O2P determined from the local positions depicted in (d). (f) Nano-FTIR absorption spectra correlated to (e) and corresponding to positions in (d).

11. In FEM simulations, clarify how the surface charge density values were determined and whether they vary with RH in the model.

Response:

Thank you for pointing this out. The reported surface charge density, σ , was measured at room temperature under ambient humidity. In our trend analysis we varied only one parameter, the dielectric constant, while holding other factors constant. We agree that σ may depend on humidity, and we plan a systematic RH-dependent study as future work.

12. Include a main-text figure comparing your device performance (voltage, current, power density) to leading MOF-based and non-MOF TENGs.

Response:

We appreciate the reviewer's feedback. Tables S3 to S5 in the revised SI summarise MOF-based TENGs using adhesion and direct-growth methods. A broad comparison with non-MOF TENGs is not very informative because device architectures, fabrication routes, polymer matrices, filler loadings, and test conditions differ substantially, and polymer composites can strongly boost output. Our aim here is to demonstrate a fabrication strategy for a pristine MOF tribopositive layer. Within the set of pristine MOF TENGs, the performance is comparable.

13. For capacitor charging experiments, calculate the mechanical-to-electrical energy conversion efficiency.

Response:

Thanks for the suggestion. The calculation for 0.1 μF capacitor was shown below. Based on the calculation the efficiency was quite low, as expected for TENG-based harvesters.

$$E_{\text{cycle}} = FS = 90 \times 0.0015 = 0.135 \text{ J}$$

$$E_{\text{total}} = 50 \times 0.135 = 6.75 \text{ J}$$

$$E_{\text{cap}} = 0.5 \times CV^2 = 0.5 \times (1.0 \times 10^{-7}) \times (2.5^2) = 3.125 \times 10^{-7} \text{ J}$$

$$\eta = E_{\text{cap}} / E_{\text{total}} = 6.753.125 \times 10^{-7} \approx 4.6 \times 10^{-8}$$

14. Discuss optimal load matching for maximum power transfer and how this compares with real application scenarios.

Response:

Thanks for the suggestion. TENG is a high-impedance and mostly capacitive source. In the lab scale TENG performance, the peak of power ($P=V^2/R$) is always the sweet spot load with optimal load. However, in the real application, the fixed resistor is not used, instead, a rectifier plus a charging capacitor and electronics. Thus, the effective load changes over time as the capacitor fills and there are switch losses, so the sweet spot is not a fixed value here. In order to get a real application, the fixed-resistor R can be a benchmark of the harvester, while maximum energy delivery in practice comes from keeping the rectifier input near that sweet

spot with a high-impedance front end and simple power-management, such as a tiny boost that keeps the rectifier input in a good range and the real-world metric, such as how fast you charge a capacitor to a target voltage and the overall efficiency.

The revision can be found on Page 12.

15. Have XPS or related analyses been performed before and after high-RH exposure to check for chemical changes?

Response:

In this study, we conducted ATR-FTIR measurements to track changes in infrared absorption bands before and after the humidity test (approximately 3 h). The spectra show an increase in the relative intensity between the peaks at 2923 cm^{-1} and 1374 cm^{-1} , see new Figure S16 in the revised SI.

The revised text can be found on Page 17:

After the humidity stability test, FTIR spectra collected before and after humidity exposure (Figure S16) revealed a marked change in the relative peak intensity of the 2923 cm^{-1} and 1374 cm^{-1} bands, corresponding to the C-H and C-O vibrational modes. Particularly, the relative increase in the intensity of the 2923 cm^{-1} band, assigned to C-H stretching of the BTC linker can be attributed to water adsorption within the HKUST-1 pores.

16. Explain whether the performance is strongly dependent on using copper as the substrate; could other conductive substrates be used?

Response:

Electrodeposition was used throughout. The copper substrate served as the anode and was oxidised to supply Cu^{2+} for HKUST-1 formation. If a different metal were used as the anode, for example Zn or Co, related BTC-based frameworks could form instead. Changing the cathode material mainly affects proton reduction and local electrochemistry; this can influence film quality and adhesion but does not provide the metal ions for the framework. Accordingly, the present process is copper dependent. Exploring other metals is a promising direction for future exploration.

The revised sentence was added on page 7:

The electrosynthesis occurs on the anodic side, where the copper metal was used as a metal source and was oxidised to copper ions (Cu^{2+}), dissolving into the electrolyte (eq. 2).

17. Provide direct or literature-based quantitative values for Young's modulus and hardness of (222) vs. other facets to support mechanical stability claims.

Response:

Thank you for the helpful suggestion. As noted in the main text, Z. Zeng *et al.* (Communications Chemistry, 2023) performed *in situ* compression of HKUST-1 along the (111) and (100) directions. They reported Young's moduli with the ratio $E(111) \approx 3.6 \times E(100)$ and yield strengths $Y(111) \approx 2 \times Y(100)$, indicating that the (111) facet is more resistant to elastic compression. These trends align with DFT calculations of the single crystal elastic constants for HKUST-1, with $E(100) \approx 3$ GPa and $E(111) \approx 15$ GPa. The hardness values were $H(100) = 463 \pm 58$ MPa and $H(111) = 491 \pm 28$ MPa, so (111) is about 1.06 times harder than (100). Because (111) and (100) correspond to the two most intense peaks in the simulated XRD, we focus on this pair; other facets will be examined in future work.

Revision can be found on Page 13:

Additionally, compared to the (200) facet, the (222) facet demonstrated a relatively higher mechanical properties, specifically in hardness (yield strength), $Y_{(111)} \approx 2 \times Y_{(100)}$ and Young's modulus (stiffness), $E_{(111)} \approx 3.6 \times E_{(100)}$.

18. In long-term cycling tests, provide statistical analysis of any subtle decay trends over 97,000+ cycles.

Response:

The output declined from 95 V to 93 V (1.9% decrease). The decay is non-monotonic with occasional fluctuations, plausibly due to environmental factors and layer saturation.

Statistical analysis for Figure 3a appears on Pages 10-11: **Over 13.5 h of continuous operation, we found that the TENG device maintained a stable output of 93.6 ± 1.8 V, with only a decline of 1.9%.**

Also amended in Figure 3's caption: **(exponential decay constant k was estimated in eqn. $\ln(95.4/93.6)/97000$, thus giving $k = 1.99 \times 10^{-7}$ cycle⁻¹)**

19. Discuss potential sensitivity to dust/organic contaminants on the HKUST-1 surface and how this might affect real-world operation.

Response:

We appreciate the reviewer's suggestions. Device sensitivity can be affected if the surface is exposed to dust or organic contaminants. Such contamination can physically block pores, limiting charge storage, altering the dielectric properties, and shortening the charge lifetime. Mechanical properties (e.g., adhesion) and effective surface area are also impacted by a dust overlayer, resulting in a reduced real contact area. Certain organic contaminants may chemically modify HKUST-1 (for example, by changing its polarity), which can alter its triboelectric behaviour and shift its position in the triboelectric series. For real-world operation, protective measures should be implemented, such as encapsulating the device in a dust-free enclosure, applying a very thin protective coating, implementing a washing procedure for reuse, or assembling a sealed sandwich structure to enhance the contact area and reduce variability in one of the HKUST-1 layers.

Revisions to the text can be found on Page 20.

20. Expand the discussion on where such MOF-based TENGs could be uniquely impactful compared to other tribopositive materials (e.g., in marine sensors, wearable electronics).

Response:

Thanks for the suggestion. Among tribopositive materials, HKUST-1 combines open Cu^{2+} sites and hydrophilic, crystalline pores with facet-addressable surfaces that can be patterned by electrodeposition directly on copper substrate. In humid coastal air (≈ 70 RH%), mild water uptake at the open Cu sites sustains interfacial polarization without excessive leakage, allowing reliable powering of humidity or wave-motion nodes on copper, but the corrosion of copper should be taken into account. For wearables, electrodeposited HKUST-1 films can be integrated into textiles or flexible foils so they bend with the fabric; the copper framework can be tuned for skin contact and may offer antimicrobial benefits via releasing of low-level Cu^{2+} . While breath or sweat modulates the signal so the same device doubles as a humidity monitor. For chemical sensing, HKUST-1 selectively adsorbs gases (CO_2 , NH_3 , alcohols, N_2); when these are present, the TENG output changes or even direct colour change, enabling air-quality or leak detection without extra power. (222) Facet control on micro-textured copper further supports compact, efficient harvesters for infrastructure and IoT nodes.

The revision is given on Page 20:

HKUST-1, with open Cu^{2+} sites and hydrophilic pores, can be electrodeposited and patterned on a solid copper substrate to yield a well faceted polycrystalline film. At $\sim 70\%$ RH, modest water uptake sustains interfacial polarisation with low charge leakage, enabling reliable operation of wearables and portable devices. Although sweat or breath can modulate the signal for self-powered humidity sensing, the corrosion of copper should be considered for practical implementations. Selective adsorption of CO_2 , NH_3 , alcohols and N_2 in HKUST-1 could also change its output or the film colour, in which the

crystallographic control of the (222)-oriented facets on micro-textured copper may boost efficiency for integration in compact harvesters targeting internet of things (IoT).

I recommend major revision. The work is promising and could make a valuable contribution to the TENG and MOF materials field, but addressing these queries with additional data, analysis, and clarifications is essential to meet the rigor and impact expectations.

Response:

Thanks for the recommendation. We have taken your suggestions and improved the quality of the revised manuscript.

Department of Engineering Science
University of Oxford
Parks Road
Oxford OX1 3PJ
United Kingdom

Prof. Jin-Chong Tan
Professor of Nanoscale Engineering
Fellow of Balliol College, Oxford.
Telephone: +44 (0)1865 273925
e-mail: jin-chong.tan@eng.ox.ac.uk
URL: <https://eng.ox.ac.uk/mmclab>

2 December 2025

Dear Editor,

Manuscript ID: COMMSCHEM-25-0632A.R2

Following the referee's additional feedback, we have performed new experiments and analyses as described below. Amendments to the revised manuscript and additions to the Supplemental Information (SI) are **highlighted** within.

Reviewer #1 (Remarks to the Author):

The authors have well answered the questions. I thus recommend the publication of this paper.

Reviewer #2 (Remarks to the Author):

The revised manuscript, "Robust triboelectric energy harvesters engineered from electrochemically deposited films of HKUST-1 polycrystals," has partially addressed the comments. Still the novelty of the manuscript is not up to the mark to publish in such prestigious journal like Nature Communications Chemistry. A few trials can achieve the direct growth of MOFs on conducting substrates by electrodeposition, and the manuscript does not discuss what the thumb rules/guidelines are for electrodeposition of MOFs on conducting substrates in general sense. Further, the application part is so weak, as per the journal's broad audience, it should be a new potential application. Based on TENG performance, it is not particularly high-performance. Based on the above, I strongly recommend a rejection. However, I am giving clearer comments to improve the manuscript.

1. There is no reason why the author chose Kapton as the opposite layer. As the author expected a positive nature of the HKUST, in that they should have chosen more tribonegative films like (FEP, PDMS, Silicone) as an opposite to get better TENG performance.

Response:

Thank you for your valuable comment. We agree that tribonegative materials such as FEP, PDMS, or silicone could potentially enhance the TENG performance when paired with a

positively charged material like HKUST. Kapton was selected in this study for several practical and technical reasons. First, it is a commercially available material that can be easily obtained at a reasonable cost. Second, Kapton exhibits excellent chemical and thermal stability, maintaining its integrity up to 260 °C without melting or degradation, and it is resistant to most solvents, oils, and chemicals. In addition, it shows good electrical stability and UV resistance. Moreover, Kapton has lower adhesion compared to PDMS or FEP, which helps minimize the unwanted transfer of HKUST crystals onto the tribonegative surface during operation. Actually, we have attempted the use of PDMS, it was unstable due to its higher surface adhesion. Nevertheless, we acknowledge that exploring other tribonegative films may further improve the output performance, and we plan to investigate this in future work.

2. Authors should provide real-time testing videos for TENG testing under different humidity levels and powering LEDs.

Response:

Thanks for the suggestion. We have included a real-time testing video demonstrating the TENG operation and LED powering. As the 2h-HKUST-1 TENG shows a stable voltage response under varying humidity conditions, the LEDs are expected to exhibit similar brightness to that observed under normal conditions (approximately 20 °C and 50% RH), as shown in the Movie Clip in SI.

The updated revision found on page S25:

Movie Clip

Movie S1: Real-time testing of the 2 h-HKUST-1 TENG for illuminating 48 LEDs in the dark, under ambient conditions.

3. The author should provide photographs of the HKUST film before and after stability of 10 K cycles.

Response: The photographs of the HKUST-1 film before and after the stability test of 10,000 cycles are provided below. From the overall observation, the film remains stable and shows no obvious changes at the macroscale.

The revision can be found on page S11:

Photographs taken before and after the stability test are provided in Figure S9e.

Figure S9. e) Photos taken before and after ~10k cycles tested in contact-separation motion. The color change was due to the uptake of moisture.

4. Author should make a comparison table of the performance of direct growth MOF-based TENGs in the main manuscript (ex: ZIF-67, ZIF-8, MOF-303 etc)

Response:

We have presented the comparison table in supporting information, see Table S3 (directly grown MOF-based TENGs), which is a more appropriate location than the main manuscript. This choice of placement helps to maintain clarity and flow of the main manuscript, as the table contains extensive data and references that will otherwise detract from the main discussion.

5. The author should provide the test set-up used for TENG testing (photo/video), as the authors used a different set-up compared to the typically used linear motors.

Response:

Thank you for your suggestion. The photograph of the test setup under contact-separation mode, with annotations, is provided below, see Figure S1b on page S2 of SI.

Figure S1.b) Photograph of TENG set-up under contact-separation mode.

6. Authors should provide force, area, frequency dependent studies to have complete picture of HKUST based TENGs.

Response:

Thank you for your comments. We agree that force-, area-, and frequency-dependent studies are important for understanding the performance of HKUST-1-based TENGs. In our setup, the applied force is closely related to the distance between the two tribomaterials; thus, varying the distance inherently changes the contact force. In this work, we employed a standardized testing condition (see Method section), as our primary aim was to investigate the effects of growth

time and nanoscale crystal characteristics on the output performance. In fact, the suggested parametric study is systematic work pertinent only to yield future device scale up and optimisation outside scope of the current work.

7. Even though the authors used a 100X probe, the oscilloscope still gets loaded due to impedance mismatch, which leads to an error in measuring the output voltage. Justify.

Response:

Thank you for your insightful comment. To mitigate this issue, a 100× high-impedance probe was employed, which substantially increases the input impedance of the measurement system and effectively minimizes the loading effect. Although achieving complete impedance matching is challenging due to the intrinsically high internal resistance of TENGs, all measurements in this study were conducted under identical conditions. Therefore, the relative comparisons and the conclusions drawn from the experimental results remain valid and unaffected by this limitation.

8. How did the authors measure current with an electrometer? Current measurements with SR 570 give a more accurate value of currents.

Response:

In our current measurements, we selected an appropriate range in the microampere scale and properly grounded the shielding to minimize induced noise and charge accumulation. The short-circuit current was recorded using the Keithley 6514 electrometer. As the Keithley 6514 measures current directly through a built-in ultra-high-impedance feedback resistor, it provides high precision and low noise for DC and low-frequency measurements. While the SR570 current preamplifier offers higher bandwidth and is advantageous for capturing fast transient signals, the electrometer was chosen for its superior accuracy, stability, and suitability for steady-state and periodic TENG measurements in this study.

9. Authors should provide charge transfer measurements and the efficiency of the present TENG.

Response:

In the manuscript, we have included charge transfer measurements showing a maximum transferred charge of 9.8 ± 0.3 nC (corresponding to a surface charge density of approximately $11 \mu\text{C m}^{-2}$), a closed-circuit current of 17.8 ± 0.2 μA (current density of about 20 mA m^{-2}), and a power density of 771.8 ± 0.3 mW m^{-2} . These results have been standardized to allow a direct comparison with previously reported TENG performances in the literature. In future work, we plan to perform additional charge transfer measurements and further evaluate the overall energy conversion efficiency of the TENG.

10. Few cycles from the stability can be shown in Zoom.

Response:

Thank you for pointing this out. The representative cycles from the beginning and end of the stability test are shown below.

The revised text can be found on pages 11 and 12, with Figures 3a, S9a and S9b in SI:

Magnified view of a few representative cycles is shown in Figures 3a, S9a and S9b, highlighting the detailed features of each contact-separation peak.

Figure 3. Electrical performance of 2-h HKUST-1 TENG. a) Durability test of the 2 h HKUST-1 sample over 13.5 h (97,000 continuous impact cycles), with zoomed-in views of the initial and final cycles.

Figure S9. a) Representative cycles at the beginning and b) end of stability test.

11. Authors should provide work function data of both the frictional layer (HKUST and PET) to get clarity on charge transfer and the triboelectric nature of the HKUST.

Response:

Thank you for your advice. We agree that the work function data of both frictional layers are important for understanding the charge transfer mechanism and clarifying the triboelectric nature of HKUST. We have conducted KPFM measurements on HKUST-1 samples with different growth times. Since the Kapton layer remains identical across all devices, the observed variations in the surface potential of HKUST-1 indicate differences in their surface properties, which contribute to the performance variations among the TENGs. Regarding the PET layer, it serves primarily as a conductive substrate to support the Kapton film.

Reviewer #3 (Remarks to the Author):

The authors have made substantial and thoughtful revisions in response to previous reviews. The updated data, added figures (e.g., Figures S8, S11, S13, S15–S16), and clarifications in methodology significantly strengthen the manuscript. The work presents a clear demonstration of a directly electrodeposited HKUST-1 film as a tribopositive layer with stable TENG performance under high humidity, which has both scientific merit and practical relevance.

The paper is now close to publication, pending minor clarifications and small improvements as listed below.

1. The discussion (page 19) explains the interplay between dielectric constant, water adsorption, and charge screening at varying RH. → Could the authors include a concise table or plot summarizing normalized voltage versus humidity for the optimal 2 h sample to facilitate direct comparison across RH values (10–70 %)?

Response:

Thanks for your advice. The revision can be found on page 16 and Figure S15:

Specially for 2 h-HKUST-1 sample, the normalized voltages and associated errors were obtained by dividing each measured value by the maximum voltage (93.5 V). As shown in Figure S15, the voltage did not exhibit a strictly consistent increase with decreasing RH. Minor fluctuations were observed around 40 RH%, which can be attributed to environmental factors such as slight temperature drifts, transient humidity instability, and adsorption–desorption lag effects.

Figure S15. Normalised voltage to the maximum voltage versus RH (10 - 70 RH%) for the 2 h-HKUST-1 sample.

2. Figure 3b and Figure S9a–b show pre-/post-cycling SEM images. → Please specify whether the observed microcracks (page 11) affect triboelectric output reproducibility over multiple batches, or if the output variance (± 1.6 V) in Figure S8 already accounts for such morphological changes.

Response:

Thank you for pointing this out. The output variance shown in Figure S8 already includes the effects of morphological changes such as those observed in the post-cycling SEM images. As shown in Figures 3b and S9a–b, the microcracks typically appear at the corners of crystal facets or divide facets into smaller regions without causing any significant detachment or loss of bulk material. Such minor surface cracks have only a limited influence on triboelectric output, as confirmed by repeated measurements using the same samples for over 10,000 contact–separation cycles. The slight variation in output is likely due to partial redistribution of surface charges. While electrons are primarily accumulated on the flat facets of the HKUST-1 surface, the formation of microcracks allows a small fraction of charges to migrate to the inner surfaces where transfer occurs more slowly. This delayed charge transfer results in a small but stable variance (± 1.6 V) rather than a significant loss of performance.

3. Page 13–14 highlights the dominance of the (222) facet and references prior mechanical data ($E(111) \approx 3.6 \times E(100)$). → Would the authors consider adding a short sentence quantifying how this stiffness difference might translate to expected contact area or charge density variation (qualitative scaling estimate)?

Response:

Thanks for pointing this out. Although the (111) facet has a higher elastic modulus ($E(111) \approx 3.6 \times E(100)$), which would normally reduce the real contact area under the same load

according to elastic contact mechanics, the flat and smooth surface of the (111) crystal plane of HKUST-1 largely determines the actual contact behaviour. We reasoned that this surface flatness helps to establish contact with the opposing layer, offsetting the reduced deformation ability of the stiffer facet. In addition, as shown in Figure S13, the surface potential is related to the surface area, indicating that with a larger contact area, the output scales with the combined effect of mechanical stiffness and surface flatness rather than by stiffness alone.

The revised version was highlighted in main text on page 13:

Together with the increase of the (222) surface area, we reasoned that the higher stiffness of the (111)-oriented crystal planes³⁵ of HKUST-1 may influence the interfacial contact behaviour. Being relatively harder and stiffer, the (222) facets thereby facilitate an improved contact with the opposing layer that is more pliant. The triboelectric output hence is dependent on the contact surface mechanics; this is a function of actual contact area of the interface in dynamic motion.

4. The manuscript discusses hydrolysis of Cu–carboxylate bonds above 80 % RH and 100 °C (page 20).→ Could the authors comment briefly on the expected recovery or regeneration strategy (e.g., mild heating or vacuum drying) for restoring performance after extended humid exposure?

Response:

Thanks for your advice. The revise text was added on page 21 as below:

The performance degradation caused by the high humidity or temperature can be recovered through mild heating or vacuum drying, which removes adsorbed water and restores the coordination structure of the Cu-carboxylate bonds.

5. Several supplemental tables (S3–S7) and figures are cited.→ Please ensure all new supplemental materials are sequentially numbered and cross-referenced consistently in both the main text and SI, as minor mismatches were noted between figure numbers in text (e.g., S11/S13 renaming).

Response:

Thank you for pointing this out. We have carefully reviewed the text and find no inconsistencies between Figures S11, S13, or the other related figures.

Department of Engineering Science
University of Oxford
Parks Road
Oxford OX1 3PJ
United Kingdom

Prof. Jin-Chong Tan
Professor of Nanoscale Engineering
Fellow of Balliol College, Oxford.
Telephone: +44 (0)1865 273925
e-mail: jin-chong.tan@eng.ox.ac.uk
URL: <https://eng.ox.ac.uk/mmclab>

1st February 2026

Dear Editor,

Final revisions for manuscript COMMSCHEM-25-0632B

Following the referee's additional feedback, we have amended the manuscript and SI as **highlighted** within. Also we have implemented editorial changes requested to comply with your journal policies and formatting style.

Reviewer #2 (Remarks to the Author):

The revision is not satisfactory. Most reviewer comments have been omitted or addressed only with brief logical explanations, without performing the necessary additional experiments. For example, the inclusion of a comparison table is essential—since the authors claim a TENG based on direct growth of MOFs, such comparative data is valuable, especially given the limited literature in this area. Several other important comments also remain inadequately addressed.

Response:

Thank you very much for the reviewer's comment. We would like to clarify that the comparison table of TENGs based on the direct growth of MOFs has already been included in Supplementary Table 3.

Considering the limited number of reported works in this specific area, this table offers a meaningful and relevant comparison in the state of the art in the field of MOF-based triboelectrics. Furthermore, we have included comparisons of different MOF adhesion methods used in MOF TENG devices in Supplementary Tables 4 and 5 to further support our findings.

Regarding the other comments that were considered inadequately addressed, we have conducted all additional measurements and analyses that were feasible with the available facilities in our laboratory. Specifically, we performed detailed chemical and physical characterization at the nanoscale using nano-FTIR to better understand the material properties and interfacial behaviour; this is nanoscale characterisation of local chemical behaviour with regards to triboelectric response is unconventional in the field.

We acknowledged that certain suggested experiments, such as current measurement using an SR570 amplifier and direct measurement of charge transfer, could not be carried out at this stage. These points have been carefully considered, and are warranted for future studies either by our group or other researchers in the field.

We hope that the added comparative tables, additional analyses, and clarifications adequately address the reviewer's concerns.

Kind regards,

J-C Tan

The manuscript “ Robust triboelectric energy harvesters engineered from electrochemically deposited films of HKUST-1 polycrystals” is not having any potential novelty for the publications in Nature Communications Chemistry.

1. Novelty claimed by author (direct growth of MOFs on conducting substrates or direct growth of MOFs and utilized in TENG design) is well reported in the literature.

(ex: <https://pubs.rsc.org/en/content/articlehtml/2023/ta/d3ta03231k>,
<https://pubs.rsc.org/en/content/articlelanding/2018/nj/c7nj03171h>,
<https://pubs.acs.org/doi/full/10.1021/acssuschemeng.3c05198>,
<https://advanced.onlinelibrary.wiley.com/doi/full/10.1002/aenm.201803581>)

2. There are many TENG reports on HKUST (ex: <https://advanced.onlinelibrary.wiley.com/doi/full/10.1002/adfm.201807655>)
3. The output power of the TENG reported is small compared to the existing MOF-based TENGs.
- 4.